# Durotaxis is a driver and potential therapeutic target in lung fibrosis and metastatic pancreatic cancer

Taslim A. Al-Hilal[1,2,3,4,21], Maria-Anna Chrysovergi [1,2,3,21], Paula E. Grasberger[1,2,3], Fei Liu [5], Vera Auernheimer[6], Yan Zhou[1,2,3], Zebin Xiao [7], Mark Anthony Leon-Duque [8], Alba Santos[1,2,3], Tamanna Islam[4], Matteo Ligorio[9], Delphine Sicard [10], Clemens K. Probst[1,2,3], Vladimir Vrbanac[3], Tejaswini S. Reddi [11], Ludovic Vincent[12,13], Cassandra Happe[12,13], Edward Chaum [14], Charles R. Yates[15], Kaveh Daneshvar [16], Allan C. Mullen [16], David Ting [9], Eric S. White [17], Raghu Kalluri [18], Christina M. Woo [8], Ellen Puré [7], Wolfgang H. Goldmann [6], Jose Luis Alonso[19], Andrew M. Tager[1,2,3,22], Adam J. Engler [12,13], Daniel J. Tschumperlin[10] & David Lagares [1,2,3,20] ✉

Durotaxis, cell migration along stiffness gradients, is linked to embryonic development, tissue repair and disease. Despite solid in vitro evidence, its role in vivo remains largely speculative. Here we demonstrate that durotaxis actively drives disease progression in vivo in mouse models of lung fibrosis and metastatic pancreatic cancer. In lung fibrosis, durotaxis directs fibroblast recruitment to sites of injury, where they undergo mechano-activation into scar-forming myofibroblasts. In pancreatic cancer, stiffening of the tumour microenvironment induces durotaxis of cancer cells, promoting metastatic dissemination. Mechanistically, durotaxis is mediated by focal adhesion kinase (FAK)–paxillin interaction, a mechanosensory module that links stiffness cues to transcriptional programmes via YAP signalling. To probe this genetically, we generated a FAK-FAT$^{L994E}$ knock-in mouse, which disrupts FAK–paxillin binding, blocks durotaxis and attenuates disease severity. Pharmacological inhibition of FAK–paxillin interaction with the small molecule JP-153 mimics these effects. Our findings establish durotaxis as a disease mechanism in vivo and support anti-durotactic therapy as a potential strategy for treating fibrosis and cancer.

Durotaxis is a form of directional cell migration in which cells move up gradients of matrix stiffness, independent of soluble factors (chemotaxis) or matrix-bound ligands (haptotaxis)[1]. Recent progress in developing bioengineered matrices with stiffness gradients has facilitated the study of cell durotaxis in vitro[1–6], demonstrating durotactic capacity of stem cells[3,4], cancer[7], vascular[5], epithelial[2] and immune cells[8]. These studies suggest that durotaxis may contribute to embryonic development, homeostasis and disease; however, its biological role in vivo remains largely speculative[6]. A major barrier has been the lack of high-resolution methods to measure and model spatial variations in tissue stiffness. Atomic force microscopy (AFM) has enabled nanoscale mapping of stiffness gradients in biological systems such as mouse limb bud[9], the developing *Xenopus* brain[10,11], fibrotic organs[12] and desmoplastic tumours[13,14]. These studies revealed spatiotemporal associations between stiffness gradients and cell migration. Despite these advances, the lack of genetic and pharmacological tools

**Fig. 1 | Fibrotic tissues exhibit steep stiffness gradients in mice.** Spatial mapping of matrix stiffness of healthy and fibrotic tissues obtained from mouse models of skin, lung and kidney fibrosis by in situ AFM nanoindentation. **a**, Mouse model of lung fibrosis induced by a single intratracheal (i.t.) instillation of bleomycin (1.2 U kg$^{-1}$). **b**, Picrosirius red staining (marker of fibrosis) of mouse lung tissue of saline- and bleomycin-challenged mice. Representative images are presented from $n$ = 6 mice per group. Scale bar, 100 μm. **c**, Representative 3D elastographs of saline- and bleomycin-challenged mouse lung parenchyma from $n$ = 6 mice per group. Three-dimensional stiffness maps were obtained from lung tissues in the respective regions of interest identified in **b**. The colour bar indicates Young's modulus, with red colour indicating areas of increased stiffness. **d**, Measurement of matrix stiffness as a function of distance in mouse lung tissues. Calculation of stiffness gradients and average slope based on 50 slopes per mouse lung tissue sample from $n$ = 6 mice per group. **e**, Mouse model of skin fibrosis induced by daily subcutaneous (s.c.) injection of bleomycin (0.05 U kg$^{-1}$). **f**, Picrosirius red staining of mouse skin tissues from $n$ = 6 mice per group. Scale bar, 100 μm. **g**, Representative 3D elastographs of saline- and bleomycin-challenged mouse skin from $n$ = 6 mice per group. **h**, Calculation of stiffness gradients and average slope based on 50 slopes per mouse skin tissue sample from $n$ = 6 mice per group. **i**, Mouse model of kidney fibrosis induced by unilateral ureteral obstruction (UUO). **j**, Picrosirius red staining of mouse kidney tissues from $n$ = 6 mice per group. Scale bar, 100 μm. **k**, Representative 3D elastographs of UUO- and sham-operated mouse kidneys from $n$ = 6 mice per group. **l**, Calculation of stiffness gradients and average slope based on 50 slopes per mouse kidney tissue sample from $n$ = 6 mice per group.

to target durotaxis-specific pathways in vivo has limited mechanistic studies in these models. Recent work has identified molecular pathways involved in detecting stiffness gradients[15], a process known as mechanosensing, largely controlled by integrins and focal adhesion-associated proteins[16]. Notably, these pathways underlying durotaxis appear to be dispensable for chemotaxis or haptotaxis[17]. Here, we identify the focal adhesion kinase (FAK)–paxillin interaction as a durotaxis-specific mechanosensory module and demonstrate that its genetic or pharmacological disruption reduces disease progression in mouse models of lung fibrosis and metastatic pancreatic cancer in vivo.

## Results

### Fibrotic tissues exhibit steep stiffness gradients

The pathological recruitment of fibroblasts to sites of tissue injury and their subsequent activation into scar-forming myofibroblasts are key steps in the development and progression of organ fibrosis[18]. While fibroblast recruitment via chemotaxis is a well-established disease mechanism in tissue fibrogenesis[19,20], the role of durotaxis remains poorly understood. We hypothesized that fibrogenic injury

generates local stiffness gradients sufficient to guide fibroblast migration via durotaxis, thereby amplifying fibrotic remodelling. To investigate the local spatial distribution of matrix stiffness with nanoscale precision, we applied in situ AFM nanoindentation concurrent with post-hoc image co-registration and picrosirius red staining (marker of fibrosis) in healthy and fibrotic tissues from three mouse models of organ fibrosis: lung, skin and kidneys (Fig. 1a,e,i). Consistent with our previous work[12,21], fibrotic tissues showed elevated collagen content and increased matrix stiffness compared with healthy controls, confirmed by histological, biochemical and biomechanical analyses (Fig. 1b,f,j and Extended Data Fig. 1). Striking spatial differences in tissue stiffness were observed between fibrotic and healthy tissues (Fig. 1c,g,k). Three-dimensional (3D) stiffness maps (elastographs) revealed that uninjured organs displayed relatively uniform stiffness with minimal spatial gradients (average slopes: lung 47 Pa μm$^{-1}$, skin 69 Pa μm$^{-1}$ and kidney 29 Pa μm$^{-1}$). By contrast, fibrotic tissues exhibit dramatic stiffness heterogeneity, with focal 'peaks' (up to 40 kPa) and 'valleys' (down to 0.5 kPa), forming steep spatial stiffness gradients from soft surrounding areas into fibrotic

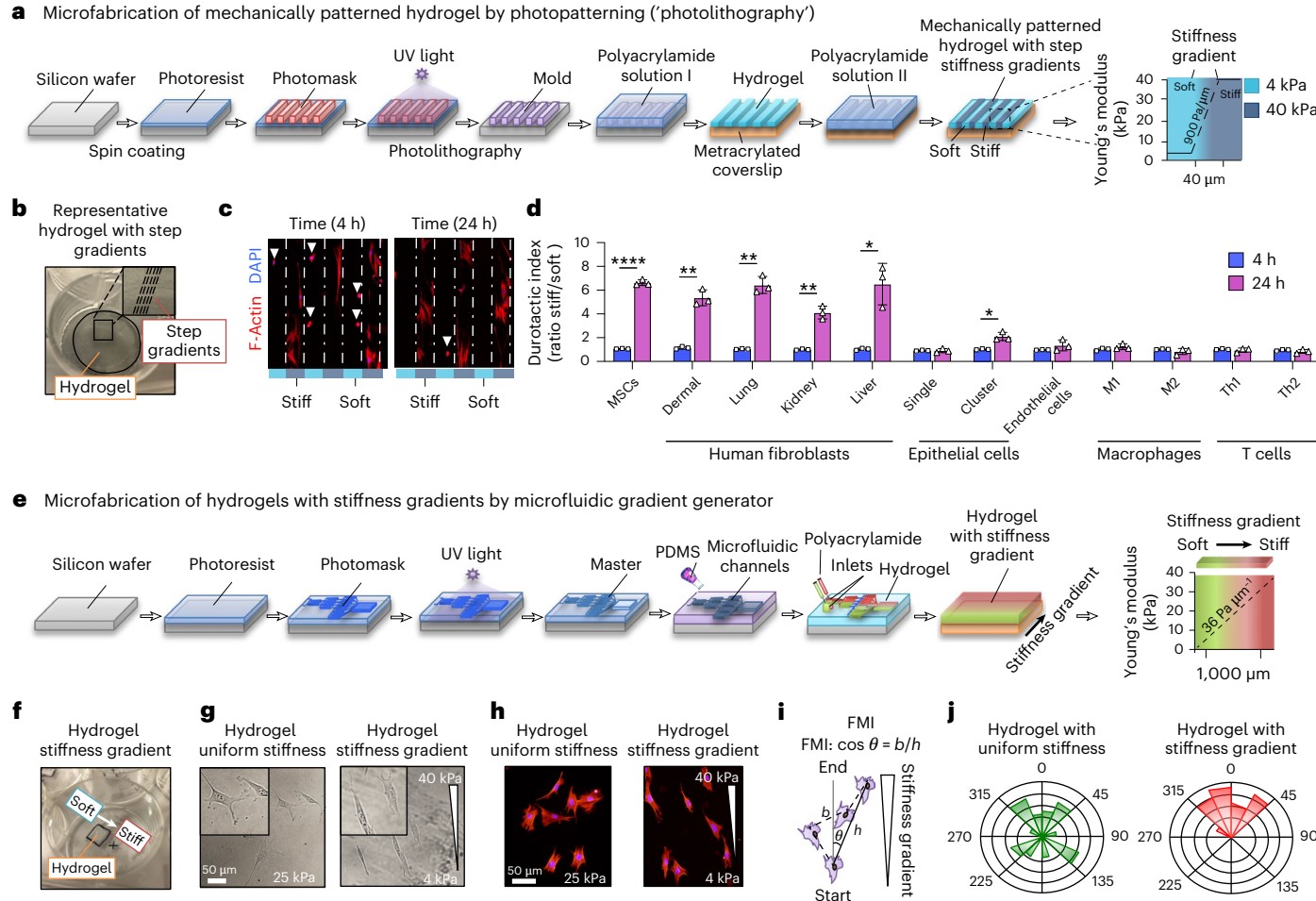

**Fig. 2 | Stiffness gradients induce durotaxis of fibroblasts. a,** A schematic of the microfabrication of mechanically patterned hydrogels to assess durotaxis. A soft polyacrylamide (PA) hydrogel (4 kPa) was initially photo-polymerized on top of a methacrylate-treated coverslip using UV light. A second PA stiff hydrogel (40 kPa) was added on top of the soft hydrogel by photo-crosslinking the polymer solution in the form of stripes using a photomask. **b,** The final product results in a 'step' hydrogel consisting of alternating soft and stiff bars, creating a stiffness gradient of 900 Pa μm⁻¹ between adjacent stripes bars that expands over 40 μm. **c,** Durotaxis assay on 'step' hydrogels. Immunofluorescence showed that fibroblasts durotax to stiff bars 24 h after plating. Fibroblasts were identified by staining for phalloidin (red) to visualize F-actin and DAPI (blue) to visualize nuclei. Mitomycin C treatment was used to prevent proliferation. **d,** Durotaxis index of multiple human and murine cell types involved in tissue fibrosis. MSCs, mesenchymal stem cells. Data were obtained from three biological replicates each. Two-way ANOVA test. *$P < 0.05$, **$P < 0.01$, ***$P < 0.001$ versus 4 h.

**e,** A schematic of the microfabrication of hydrogels with continuous stiffness gradients by microfluidic gradient generator. As shown in **f,** the final product results in a hydrogel in which the stiffness gradient (36 Pa μm⁻¹) incorporates an identical change in matrix stiffness over a greater distance (1,000 μm). **g,h,** Representative brightfield (**g**) and immunofluorescence (**h**) images of lung fibroblasts plated either on hydrogels of uniform stiffness (25 kPa) or hydrogels with stiffness gradients (4–40 kPa). Scale bar, 25 μm. $n = 3$ independent experiments. **i,** A schematic of a cell trajectory and the variables used to calculate the forward migration index (FMI) in time-lapse imaging studies. Angular displacement was used to determine the cell's angular trajectory. **j,** Rose diagrams of cell migration showing angular distributions of cell trajectories in their migration tracks relative to the stiffness gradient, with the radius length indicating the number of events in each trajectory. Data were obtained from 4 biological replicates, $n = 68$–77 cells. For cell-based assays, data are given as mean ± s.d. from three independent experiments.

niches[12]. Importantly, the magnitude of these stiffness gradients (average slope: lung fibrosis 500 Pa μm⁻¹, skin fibrosis 433 Pa μm⁻¹ and kidney fibrosis 115 Pa μm⁻¹) falls within the range known to promote durotaxis in vitro[22] (Fig. 1d,h,l). These findings suggest that durotaxis may be an active mechanism driving fibroblast, recruitment, localization and contributing to fibrotic tissue remodelling in vivo.

## Stiffness gradients drive fibroblast durotaxis in vitro

To test whether stiffness gradients in fibrotic tissues can activate durotaxis, we used photolithography to engineer mechanically patterned polyacrylamide (PA) hydrogels with alternating 4 kPa (soft healthy tissue) to 40 kPa (stiff fibrotic tissue) stripes over 40 μm (ref. 22), mimicking in vivo conditions with a gradient of 900 Pa μm⁻¹ (Fig. 2a). Cells were plated in serum-free medium, and the durotactic index

was determined as the ratio of cells on stiff versus soft regions after 24 h (ref. 22) (Fig. 2b,c). Cell accumulation patterns reflected migration rather than differences in proliferation on stiff stripes or apoptosis on soft stripes, as described previously[22]. We then assessed the durotactic index of multiple human and murine cell types involved in fibrosis (Fig. 2d and Extended Data Fig. 2a–l). Mesenchymal stem cells (MSCs) and fibroblasts showed strong durotaxis (index >6), endothelial cells showed moderate durotaxis (index >2) and epithelial cells durotaxed only at high confluence (index >4), consistent with collective migration[2]. By contrast, innate and adaptive immune cells, including M1/M2 macrophages and T cells, showed no durotaxis. Notably, neither epithelial nor immune cells demonstrated durotaxis on hydrogels coated with laminin, ruling out substrate affinity as the reason for the lack of durotaxis in these cells (Extended Data Fig. 2m). To further investigate

fibroblast durotaxis, we engineered a second set of shallower hydrogels using microfluidics, creating the same stiffness range (4–40 kPa) over a longer range (1 mm) (Fig. 2e,f), which allowed long-term tracking of single-cell directional migration via time-lapse microscopy (Supplementary Video 1). Immunocytochemistry showed that fibroblasts developed morphological front-to-rear polarity oriented towards stiffer regions on gradients, unlike cells on uniform stiffness gels (Fig. 2g,h). Quantitative time-lapse image analysis confirmed that fibroblasts persistently move up stiffness gradients, whereas cells on uniform gels moved randomly (Fig. 2i,j). Together, these results show that fibroblasts undergo durotaxis on stiffness gradients that resemble fibrotic tissues, further supporting a role for durotaxis in tissue fibrogenesis.

## Fibroblast durotaxis is controlled by FAK–paxillin in vitro

Durotaxis begins with matrix 'mechanosensing', where integrin-based adhesions function as mechanosensors. Cells generate traction forces and interpret the resistance of the extracellular matrix (ECM) to deformations as spatial information that guides directional migration[23]. On stiffer regions, focal adhesions are mechanically reinforced through positive feedback loops. Specifically, high ECM resistance activates FAK, phosphorylates the adaptor protein paxillin at tyrosines Y31 and Y118 and promotes the recruitment of vinculin, stabilizing the adhesion complex[17,24]. This enables cells to generate traction forces necessary for movement along stiffness gradients. This mechanism, often described as the molecular clutch model, relies on the dynamic turnover of focal adhesions (that is, cycles of assembly–disassembly) to ensure effective mechanosensing and durotaxis[6], a process controlled by the FAK–paxillin$^{Y31/118}$ pathway[17] (Fig. 3a,b). Using proximity ligation assay (PLA), we confirmed that FAK interacts with phosphorylated Y31 paxillin specifically at the leading edge of durotaxing cells, precisely where stiffness sensing occurs (Fig. 3c and Extended Data Fig. 3a). To test its functional relevance, we engineered lentiviruses overexpressing paxillin mutants that prevent focal adhesion turnover, including constitutively active paxillin phosphomimetic (paxillin$^{Y31/118E}$), inactive non-phosphorylatable paxillin (paxillin$^{Y31/118F}$) or paxillin point mutant defective in vinculin binding (paxillin$^{E151Q}$). All three mutants significantly impaired fibroblast durotaxis but had no effect on chemotaxis induced by gradients of lysophosphatidic acid (LPA) in Boyden chamber chemotaxis assays or haptotaxis induced by fibronectin gradients in μ-Slide microfluidic chambers (Fig. 3d–f and Extended Data Fig. 3b). Together, our results define the FAK–paxillin$^{Y31/118}$ axis as a durotaxis-specific mechanosensing pathway, independent of chemotactic or haptotactic signalling[17].

Having established the FAK–paxillin interaction as a mechanosensory module essential for durotaxis, we next investigated whether small molecules could selectively disrupt this pathway. Structural studies have shown that paxillin's LD2 and LD4 motifs bind the focal adhesion targeting (FAT) domain of FAK[25,26] (Fig. 3b). In this context, we recently identified JP-153, a small molecule that disrupts FAK–paxillin interaction by preventing the association of paxillin LD2 and LD4 and the FAK FAT domain[27]. To examine the effects of JP-153 on fibroblast mechanosensing and durotaxis, we treated human lung fibroblasts cultured on defined soft (0.5 kPa) and stiff (64 kPa) PA hydrogels with increasing doses of JP-153. Stiff matrices activate the FAK–paxillin$^{Y31/118}$ pathway, as demonstrated by increased levels of phospho-FAK (Y397) and phospho-paxillin (Y31) (Fig. 3g). JP-153 selectively reduced phospho-paxillin (Y31) levels without affecting Y397 FAK autophosphorylation, indicating that it disrupts FAK–paxillin complexes without inhibiting FAK's enzymatic activity (Fig. 3g). These results were further supported by PLA (Extended Data Fig. 3c). In cell-based assays, JP-153 had no effect on chemotaxis in response to LPA (GPCR-dependent signalling) or platelet-derived growth factor (PDGF) (receptor tyrosine kinase-dependent pathway), nor did it inhibit haptotaxis (Fig. 3h–k). By contrast, submicromolar concentrations of JP-153 significantly impaired fibroblast durotaxis (Fig. 3l), without affecting

cell survival or proliferation (Extended Data Fig. 3d,e). Overall, JP-153 disrupts FAK–paxillin interaction and selectively inhibits fibroblast durotaxis without disrupting other cellular functions.

To investigate how durotaxis might drive fibrogenesis, we hypothesized that it guides fibroblasts to stiff fibrotic regions, where matrix stiffness promotes their activation into myofibroblasts, thus amplifying fibrosis through a feedback loop previously described by our group and others[12,16,21,28]. To test whether inhibiting durotaxis could block this activation process, we treated human lung fibroblasts with JP-153 and cultured them on mechanically patterned hydrogels with stiffness gradients. Fibroblasts migrated to stiff regions and expressed α-smooth muscle actin (α-SMA), a marker of myofibroblast differentiation, within 48 h (Fig. 3m). Pretreatment with JP-153 blocked both migration and subsequent α-SMA$^+$ myofibroblast formation, even in the presence of TGF-β (Fig. 3m,n). Posttreatment had no effect on already differentiated cells (Extended Data Fig. 3f), indicating that JP-153 selectively prevents durotaxis-dependent activation.

We further examined the temporal dynamics of fibroblast mechanosensing and activation across stiffness gradients. Immunostaining for phosphorylated paxillin showed increased number and size of paxillin-associated focal adhesions at the leading edge of durotaxing cells compared with the rear edge (Fig. 3o,p,q). Staining for YAP, a mechanosensitive transcriptional co-activator, revealed cytoplasmic localization on soft matrices and nuclear accumulation on stiff regions[29,30]. Notably, YAP nuclear accumulation began during durotaxis, suggesting real-time integration of mechanical cues into transcriptional activation via the FAK–paxillin–YAP axis. JP-153 reduced phospho-paxillin levels and prevented YAP nuclear localization (Fig. 3r,s). Together, these findings demonstrate that stiffness gradients link fibroblast migration and activation through FAK–paxillin–YAP signalling and that blocking durotaxis with JP-153 interrupts this profibrotic mechanism in vitro (Fig. 3t).

## Genetic disruption of durotaxis prevents fibrosis in vivo

To begin exploring the role of durotaxis in fibroblasts recruitment to fibrotic areas in vivo, we used Col-GFP reporter mice and coupled AFM with immunofluorescence to assess the mechanical properties of spatially restricted fibrotic areas and the position of fibroblasts and myofibroblasts within these niches. As shown in Extended Data Fig. 4a–e, GFP$^{high}$/α-SMA$^+$ myofibroblasts localized to stiff regions, while GFP$^{low}$/α-SMA$^-$ fibroblasts were found in softer areas. We next used fluorescence-activated cell sorting (FACS) to isolate both GFP$^{low}$ fibroblasts and GFP$^{high}$ myofibroblasts from the lungs of fibrotic Col-GFP reporter mice (Extended Data Fig. 4f). In vitro durotaxis assays revealed that GFP$^{low}$/α-SMA$^-$ fibroblasts showed increased durotactic capacity that GFP$^{high}$/α-SMA$^+$ myofibroblasts (Extended Data Fig. 4g). Notably, GFP$^{high}$ myofibroblasts expressed high levels of *col1a1* and *acta2* mRNA and showed limited durotactic capacity compared with GFP$^{low}$ fibroblasts, which expressed lower levels of *col1a1* and *acta2* mRNA and significant durotactic activity (Extended Data Fig. 4h,i), suggesting that durotaxis may precede myofibroblast activation of recruited fibroblasts in vivo. While expansion of 'fibroblast foci' in patients with idiopathic pulmonary fibrosis has been attributed to myofibroblast proliferation, our data challenge this view. In bleomycin-induced fibrosis model, less than 1% of GFP$^{high}$/α-SMA$^+$ myofibroblasts were BrdU positive, whereas over 6% of GFP$^{low}$/α-SMA$^-$ fibroblasts were actively proliferating (Extended Data Fig. 4j,k). These observations support a model in which fibroblasts proliferate, migrate via durotaxis to stiffer fibrotic regions and subsequently differentiate into activated myofibroblasts, contributing to fibroblast foci expansion.

To directly assess the role of fibroblast durotaxis in lung fibrosis development in vivo, we leveraged a genetic strategy to inhibit durotaxis by selectively disrupting the FAK–paxillin pathway. X-ray crystallography and nuclear magnetic resonance have previously shown that residue L994 in helix 3 of the FAK FAT domain is essential for

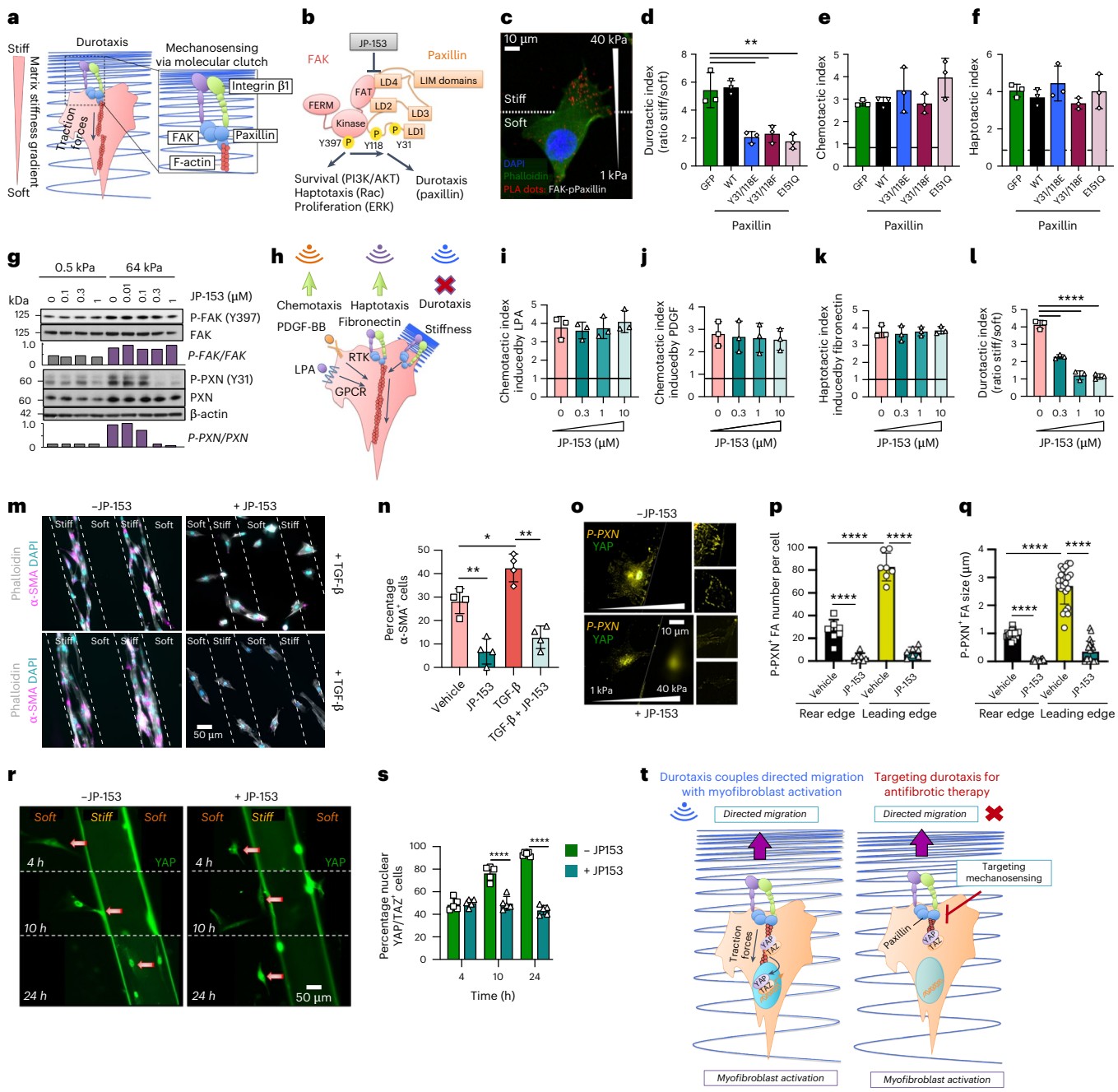

**Fig. 3 | The FAK–paxillin mechanosensitive pathway controls fibroblast durotaxis. a**, A schematic of matrix mechanosensing via the FAK–paxillin[Y31/118] pathway and its role in initiating durotaxis. **b**, A schematic of FAK-FAT domain interaction with paxillin LD4 and LD2 domains that is disrupted by a small molecule inhibitor JP-153 without affecting FAK catalytic activity. **c**, PLA demonstrates activation of the FAK–paxillin[Y31/118] pathway at the leading edge of durotaxing fibroblasts. Green indicates F-actin, and blue indicates nuclei. Red dots indicate FAK–phospho-paxillin Y31 interaction. Scale bar, 10 μm. **d–f**, The effect of lentiviral overexpression of paxillin mutants including constitutively active phosphomimetic paxillin (paxillin[Y31/118E]), inactive non-phosphorylatable paxillin (paxillin[Y31/118F]) or paxillin point mutant defective in vinculin binding (paxillin[E151Q]) on fibroblast durotaxis induced by stiffness gradients (**d**), chemotaxis induced by gradients of LPA (**e**), and haptotaxis induced by gradients of fibronectin (**f**). Data are presented as fold increase over controls (dotted line). One-way ANOVA test. **$P < 0.01$ versus GFP. **g**, The effect of JP-153 on matrix stiffness-induced activation of the FAK–paxillin[Y31/118] pathway. **h–l**, Schematic of different cell migration pathways (**h**). JP-153 dose–response effects on fibroblast, chemotaxis induced by gradients of lysophosphatidic acid (LPA) (**i**) or platelet-

derived growth factor (PDGF) (**j**), haptotaxis induced by gradients of fibronectin (**k**), and durotaxis induced by stiffness gradient (**l**). Data are presented as fold increase over controls (dotted line). One-way ANOVA test. ****$P < 0.0001$ versus 0 μM. **m,n**, The effect of JP-153 on myofibroblast formation induced by stiff matrix after fibroblast durotaxis. Activated myofibroblasts were identified by staining for α-SMA (pink, a marker of myofibroblast differentiation), phalloidin (grey, to visualize F-actin) and DAPI (blue, to visualize nuclei). Scale bar, 50 μm. One-way ANOVA test. *$P < 0.05$, **$P < 0.01$ versus vehicle. **o–q**, Focal adhesions were identified by staining for phospho-paxillin (Y31) (yellow) (**o**). Scale bar, 10 μm. The effect of JP-153 on the number (**p**) and size of focal adhesions (**q**). One-way ANOVA test. ****$P < 0.0001$ versus vehicle. **r,s**, The effect of JP-153 on the cellular localization of YAP (cytoplasmic versus nuclear) was assessed by immunostaining for YAP (green) (**r**) and its quantification (**s**). Scale bar, 50 μm. One-way ANOVA test. ****$P < 0.0001$ versus control. **t**, A schematic of the stiffness gradients coupling fibroblast durotaxis and myofibroblast activation via the FAK–paxillin[Y31/118]–YAP pathway. For cell-based assays, data are given as mean ± s.d. from three independent experiments.

binding the paxillin LD2 motif[31,32] (Fig. 4a,b). Our biophysical analyses confirmed that a point mutation (L994E) in this residue fully disrupts FAT–LD2 binding, as demonstrated by a homogeneous time-resolved fluorescence assay (Fig. 4c,d). Introducing this L994E FAK mutation into human lung fibroblasts significantly impaired durotaxis in vitro (Fig. 4e), confirming its role in mechanosensing. To investigate this mechanism in vivo, we generated a FAK[L994E] knock-in (KI) mouse using CRISPR–Cas9-mediated homology-directed repair. These mice were viable and phenotypically normal (Fig. 4f and Supplementary Methods), in contrast to global FAK- or paxillin-knockout mice, which are embryonically lethal[33,34]. When subjected to bleomycin-induced skin and lung fibrosis, FAK[L994E] KI mice exhibited significantly less fibrosis, reduced body weight loss and lower ECM accumulation compared with wild-type (WT) counterparts (Fig. 4g,h and Extended Data Fig. 5a). Moreover, primary lung fibroblasts from FAK[L994E] KI mice showed impaired durotaxis (Fig. 4i,j), reinforcing that FAK–paxillin-dependent durotaxis is essential for fibroblast migration and fibrotic tissue remodelling in vivo.

## Pharmacological inhibition of durotaxis mitigates fibrosis in vivo

We next tested whether pharmacological inhibition of FAK–paxillin-mediated durotaxis by JP-153 would replicate the anti-fibrotic effects observed in our genetic studies. In the bleomycin lung fibrosis model, prophylactic treatment with JP-153 (5 mg kg⁻¹ daily) significantly reduced lung fibrosis in the lungs at day 21 after bleomycin challenge (Fig. 4k,l). Western blot analysis confirmed inhibition of the FAK–paxillin[Y31/118] pathway by JP-153 in lung tissues (Fig. 4m). Importantly, JP-153 did not alter levels of total or active TGF-β (Fig. 4n,o), alveolar-capillary barrier permeability at day 7 (Fig. 4p) or leukocyte recruitment in the bronchoalveolar lavage (BAL) fluid (Fig. 4q,r), indicating that JP-153 specifically targets durotaxis without interfering with inflammatory or TGF-β-driven pathways. Immunohistological analysis showed that JP-153 significantly reduced the number of α-SMA⁺ myofibroblasts per cluster compared with controls (Fig. 4s,t), along with a marked reduction in ECM deposition and hydroxyproline levels, a biochemical proxy for collagen accumulation (Fig. 4u,v). Importantly, therapeutic administration of JP-153 from day 10 to 21 similarly reduced established lung fibrosis in this model (Fig. 4k,l,s,u). In vitro, JP-153 inhibited durotaxis of primary fibroblast isolated from bleomycin-induced fibrotic lungs, similar to the phenotype observed in FAK[L994E] KI fibroblasts (Extended Data Fig. 5b,c). Together, these results demonstrate that genetic or pharmacological inhibition of the FAK–paxillin pathway mitigates lung fibrosis by targeting fibroblast durotaxis in vivo.

## Metastatic tumour cells exhibit durotaxis in vitro

We next used a second disease model to further investigate the biological relevance of durotaxis in vivo. Stiffness gradients have been observed at the invasive front of desmoplastic fibrotic tumours[13,14], suggesting that durotaxis may contribute to early tumour cell dissemination and metastasis[15]. While in vitro evidence supports tumour cell durotaxis[7,23], its role in metastasis in vivo remains unproven. We selected pancreatic ductal adenocarcinoma (PDAC) as a relevant model owing to its aggressive, metastatic nature and stiff fibrotic tumour microenvironment (TME), shaped by cancer-associated fibroblasts (CAFs)[14,28,35]. We developed a patient-derived xenograft mouse model by orthotopically co-injecting GFP-luciferase-expressing pancreatic cancer cell line (PDAC3[GFP/Luc]) and an mCherry-expressing CAF at a 10:90 (PDAC[GFP/Luc]:CAF) ratio in the pancreas of severely compromised immunodeficiency (SCID) mice[36] (Fig. 5a). This model recapitulates key features of PDAC, including stromal fibrosis, tumour growth and metastasis. By day 15, histological analysis showed extensive peritumoural fibrosis driven by α-SMA⁺ CAFs (Fig. 5b,c). Small clusters of GFP⁺ tumour cells (two to three cells per cluster) were observed at the tumour invasive front (TIF), colocalizing with

fibrotic regions (Fig. 5b,c). AFM confirmed steep stiffness gradients at the tumour–stroma interface (Fig. 5d,e and Extended Data Fig. 6a), implicating durotaxis in guiding cancer cell migration. To model this behaviour in vitro, we engineered mechanically patterned hydrogels with a central soft island (1 kPa) and alternating soft and stiff (1–10 kPa) stripes to mimic the tumour–stroma interface (Fig. 5f). Pancreatic tumour cell clusters preferentially migrated along stiff stripes, consistent with collective durotaxis (Fig. 5g). These findings suggest that durotaxis may play a role in early PDAC cell dissemination and metastasis.

To assess clinical relevance, we examined durotaxis behaviour and expression of the FAK–paxillin mechanosensory module across PDAC molecular subtypes: quasi-mesenchymal (QM), classical epithelial and exocrine-like, with QM being the phenotype associated with high tumour grade, metastasis and poor survival in patients with PDAC[37,38]. Only QM PDAC lines exhibited significant durotaxis in vitro; classical epithelial lines did not (Fig. 5h). Both subtypes showed similar chemotaxis, invasion and proliferation, although with differing growth rates (Fig. 5i–k). We next assessed activation of the FAK–paxillin[Y31/118] pathway. QM tumour cells displayed increased phospho-FAK and phospho-paxillin Y31, whereas classical epithelial cells lacked expression or activation of this pathway, suggesting that functional mechanosensing is restricted to QM cells (Fig. 5l). PLA studies further confirmed the presence of FAK–paxillin complexes at the leading edge of durotaxing PDAC3 tumour cells (QM phenotype) (Fig. 5m). To directly assess the role of this pathway in tumour cell durotaxis in vitro, we engineered the PDAC3 tumour line with human paxillin knockdown and overexpression of mutant Y31E/Y118F chicken paxillin (herein PDAC3[PxnY31E/Y118F]) (Fig. 5n). These cells lacked activated phospho-Y31 paxillin and exhibited fewer focal adhesions (Extended Data Fig. 6b) but showed no differences in proliferation, chemotaxis or invasive capacity compared with controls (Fig. 5o–q). However, durotaxis was significantly impaired in PDAC3[PxnY31E/Y118F] tumour cells (Fig. 5r), a phenotype recapitulated by JP-153 treatment (Extended Data Fig. 6c). Together, our data demonstrate that the FAK–paxillin[Y31/118] pathway is selectively activated in QM PDAC tumour cells and controls their durotactic behaviour.

## Genetic disruption of durotaxis suppresses metastasis in vivo

To investigate the role of tumour cell durotaxis in tumour cell dissemination and metastasis in vivo, we utilized our orthotopic PDAC xenograft model described above[36]. In this set of experiments, we compared primary tumour growth and distal metastasis between control PDAC3[GFP/Luc] and durotaxis-deficient PDAC3[GFP/Luc/PxnY31E/Y118F] cells, co-injected with profibrotic human CAFs to ensure that both experimental groups develop the same degree of stromal fibrosis (Fig. 5s). Picrosirius red staining, immunofluorescence and AFM studies confirmed peritumoural fibrosis and matrix stiffening in both cohorts (Fig. 5s–u and Extended Data Fig. 6d), indicating similar mechanical microenvironments. No differences in angiogenesis were observed (Fig. 5v and Extended Data Fig. 7a). Assessment of primary tumour growth by histology, immunofluorescence and luciferase imaging revealed no difference in tumour mass between groups at day 15 after implantation (Fig. 5s,t,w,x). By contrast, metastatic burden in the liver and gastrointestinal tract was significantly reduced in PDAC3[GFP/Luc/PxnY31E/Y118F] tumours compared with controls, as shown by luciferase imaging and histological analysis (Fig. 5w,y and Extended Data Fig. 7b). These results demonstrate that genetic disruption of tumour cell durotaxis impairs metastasis without affecting primary tumour growth, supporting durotaxis as a potential therapeutic target in pancreatic cancer.

## JP-153 inhibits durotaxis, tumour fibrosis and metastasis in vivo

To further investigate tumour cell durotaxis in vivo, we performed two-photon imaging on precision-cut slices of subcutaneous tumours

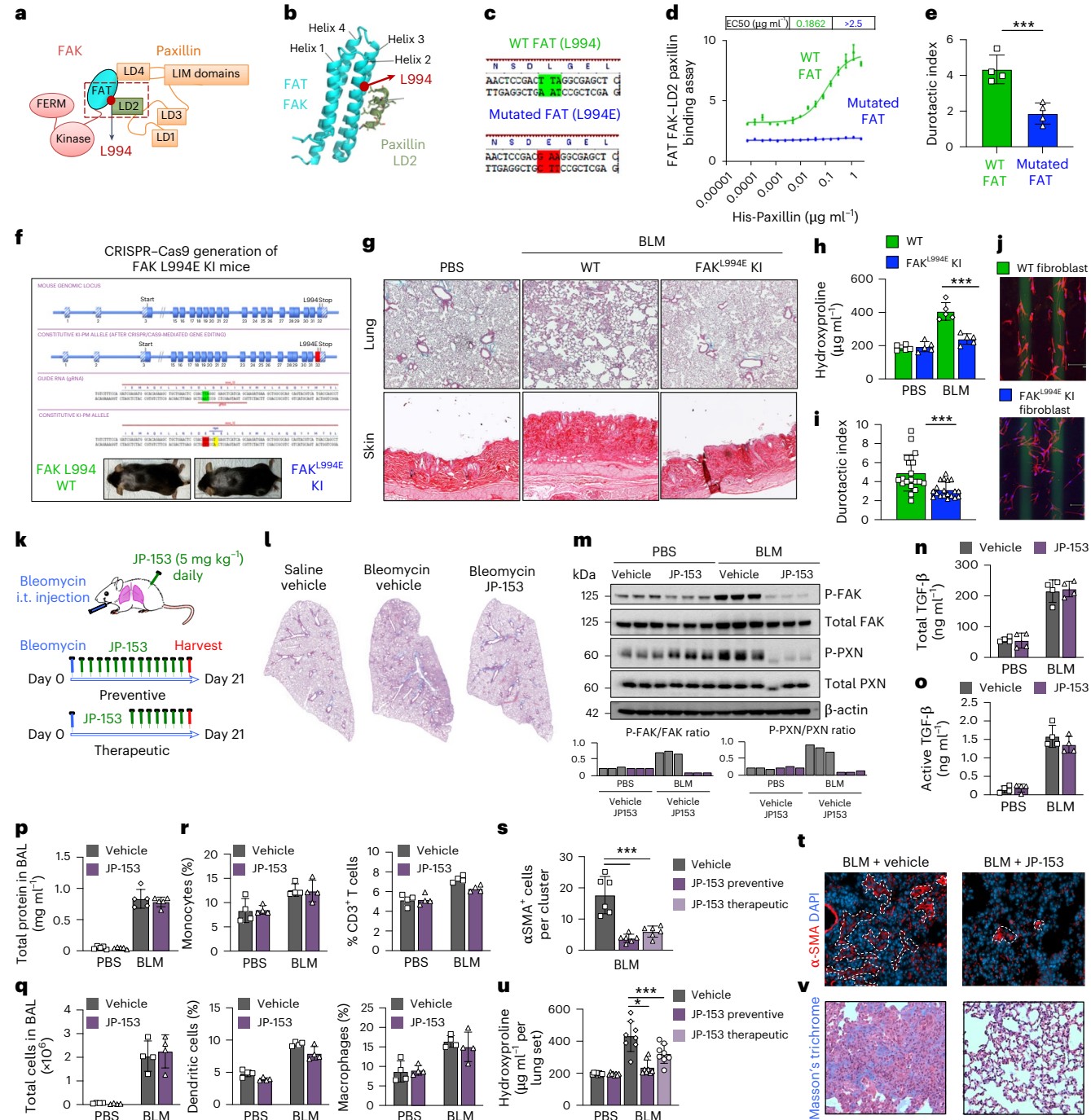

**Fig. 4 | Genetic or pharmacological inhibition of the FAK–paxillin pathway inhibits fibroblast durotaxis and organ fibrosis in mice. a**, A schematic of the interaction between FAK-FAT and paxillin LD2 domains. The red ball represents the positions of site-directed binding between FAT and LD2 domains. **b**, A 3D model of the LD2 motif of paxillin binding to FAT via L994 motif. **c**, The amino acid sequences of WT FAT on exon 32 and its point mutated version (L994E). **d**, The binding of WT FAT and mutated FAT (L994E) with paxillin LD2 domain. **e**, The effect of transfection of WT FAK and L994E FAK on fibroblast durotaxis induced by stiffness gradients. Student's *t*-test. ***P < 0.001 versus WT FAT. **f**, gRNAs targeting FAK locus used to create a FAK^L994E KI mice. **g**, H&E and picrosirius red staining of mouse lung and skin tissues from WT and FAK^L994E KI mice subjected to fibrosis models induced by bleomycin (BLM) injury. *n* = 5 mice per group. Scale bar, 100 μm. **h**, Collagen content after bleomycin treatment assessed by hydroxyproline. One-way ANOVA test. ***P < 0.001 versus WT. **i,j**, Quantification (**i**) and representative images (**j**) of durotaxis in isolated primary fibroblasts from bleomycin-treated FAK^L994E KI and WT mice with *n* = 3 for each group. Student's *t*-test. ***P < 0.001 versus WT. **k**, A schematic showing

JP-153 treatment regimens (prophylactic or therapeutic, 5 mg kg^−1 daily) in mice subjected to bleomycin-induced lung fibrosis. *n* = 6 mice for each group. **l**, Histological measure of fibrosis by Masson's trichrome stain. **m**, Representative western blot (top) and densitometry (bottom) of P-FAK, FAK, P-paxillin and paxillin protein expression levels (normalized to β-actin). One representative out of three technical replicates is shown. **n,o**, The concentration of total (**n**) and active (**o**) TGF-β protein levels, as determined by ELISA. **p**, The concentration of total protein in bronchoalveolar lavage (BAL) fluid. **q**, Percentage of total cells in BAL fluid. **r**, Percentages of different immune cells, as determined by flow cytometry, in BAL fluid at 7 days post bleomycin with *n* = 4 mice for all groups. One-way ANOVA test. ***P < 0.001 versus vehicle. **s,t**, Number of α-SMA^+ myofibroblasts (**s**) per cluster assessed by immunohistochemistry (**t**), *n* = 6 for all groups. Scale bar, 100 μm. One-way ANOVA test. *P < 0.05, ***P < 0.001 versus vehicle. **u,v**, Lung collagen content post bleomycin (**u**) and Masson's trichrome (**v**) analysis with *n* = 6 for all groups. For animal experiments, data are given as mean ± s.d.

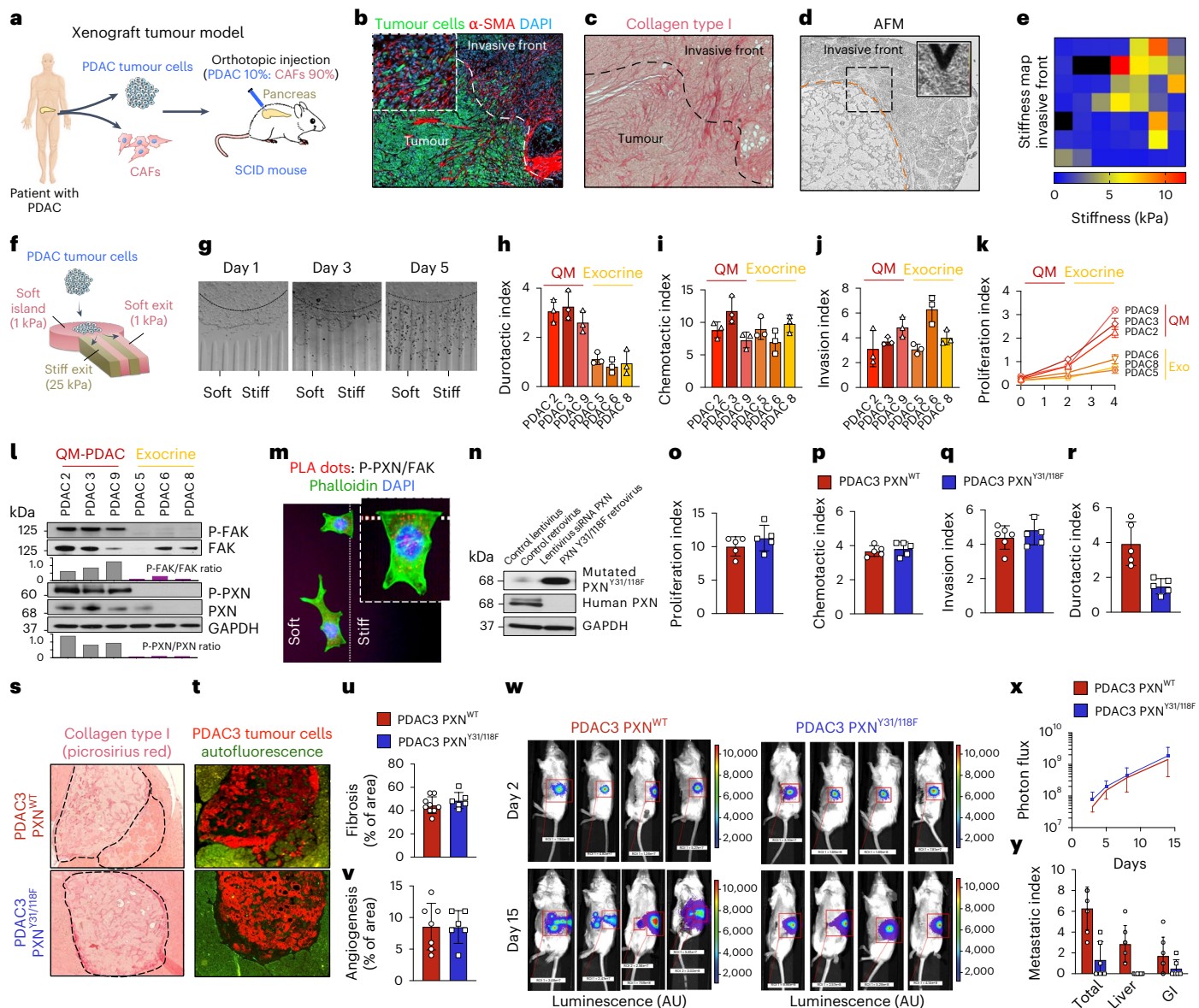

**Fig. 5 | Genetic inhibition of FAK–paxillin reduces tumour cell durotaxis and metastasis in a mouse model of pancreatic cancer. a**, A schematic of generating an orthotopic xenograft PDAC model in SCID mice. **b**, Representative immunohistochemistry showing tumour cells and CAFs at 15 days after inoculation. **c**, Detection of collagen type I and tumor fibrosis by Picrosirius red staining. **d,e**, AFM mapping (**d**) and its measurement as a function of distance (**e**) reveals increased matrix stiffness at the interface between the primary tumour and invasive front, featuring a steep stiffness gradient towards the TME (*n* = 6 for each group; scale bar, 100 μm). **f**, A schematic depicting hydrogels with mechanically patterned island of soft matrix (1 kPa) surrounded by alternating soft (1 kPa) and stiff (25 kPa) stripes used to visualize collective tumour cell durotaxis. **g**, Durotaxis assay on 'island' hydrogels. Data were obtained from three biological replicates. **h–k**, Durotactic (**h**), chemotactic (**i**), invasion (**j**) and proliferation (**k**) index of quasi-mesenchymal (QM) or classical epithelial PDAC subtypes. *n* = 3 independent tumour cell lines per subset. **l**, Representative western blot of P-FAK, FAK, P-paxillin and paxillin protein expression levels (normalized to β-actin) in QM or classical epithelial tumour cells. *n* = 3 independent tumour cell lines per subset. One of three technical replicates

shown. **m**, PLA demonstrating activated FAK–paxillin[Y31/118] signalling at the leading edge in durotaxing PDAC3 cells. Red dots indicate FAK–phospho-paxillin Y31 interactions, with F-actin (green, phalloidin) and nuclei (blue, DAPI). Scale bar, 10 μm. **n**, A representative western blot of genetically engineered control PDAC3 and PDAC3[PxnY31E/Y118F] tumour cells. *n* = 3 independent experiments. **o–r**, Proliferation (**o**), chemotactic (**p**), invasion (**q**) and durotactic (**r**) index of the WT PDAC3 tumour cell line (herein PDAC3[PxnWT]) compared with the mutated PDAC3[PxnY31E/Y118F]. **s,t**, In vivo tumorigenic assessment after orthotopic injection of PDAC3[PxnWT] and PDAC3[PxnY31E/Y118F] cells along with CAFs into the pancreas of SCID mice. Primary tumour growth and fibrosis evaluated by picrosirius red staining (**s**) and immunofluorescence (**t**) at day 15. *n* = 6 for each group. Scale bar, 100 μm. **u,v**, Quantification of fibrosis (**u**) and angiogenesis (**v**) in the primary pancreatic tumours (*n* = 6 for each group). **w**, Representative bioluminescence images of PDAC3[PxnWT] and PDAC3[PxnY31E/Y118F] tumours at days 2 and 15 after inoculation. Scale bar photon flux, luminescence (a.u.). *n* = 8 for each group. **x**, Proliferation curves of PDAC3[PxnWT] and PDAC3[PxnY31E/Y118F] tumours at days 2, 5, 9 and 15 after inoculation. **y**, Total, liver and gastrointestinal (GI) metastatic index at euthanasia. Data are given as mean ± s.d. from three independent experiments.

formed in syngeneic mice using KPC mice expressing the YFP lineage tag (KPCY) cells, derived from Pdx1-Cre; LSL-Kras[G12D/+]; LSL-Trp53[L/+]; Rosa26[YFP/YFP] mice[39] (Fig. 6a). We first examined collagen structure at the tumour core (TC) and TIF using label-free second-harmonic generation

imaging. Fibres at the TC were curly and anisotropic, while those at the TIF were thicker, more linear and aligned, features known to stiffen the ECM (Fig. 6b–e). These structural differences correlated with stiffness gradients at the TC–TIF interface, consistent with AFM data[40]

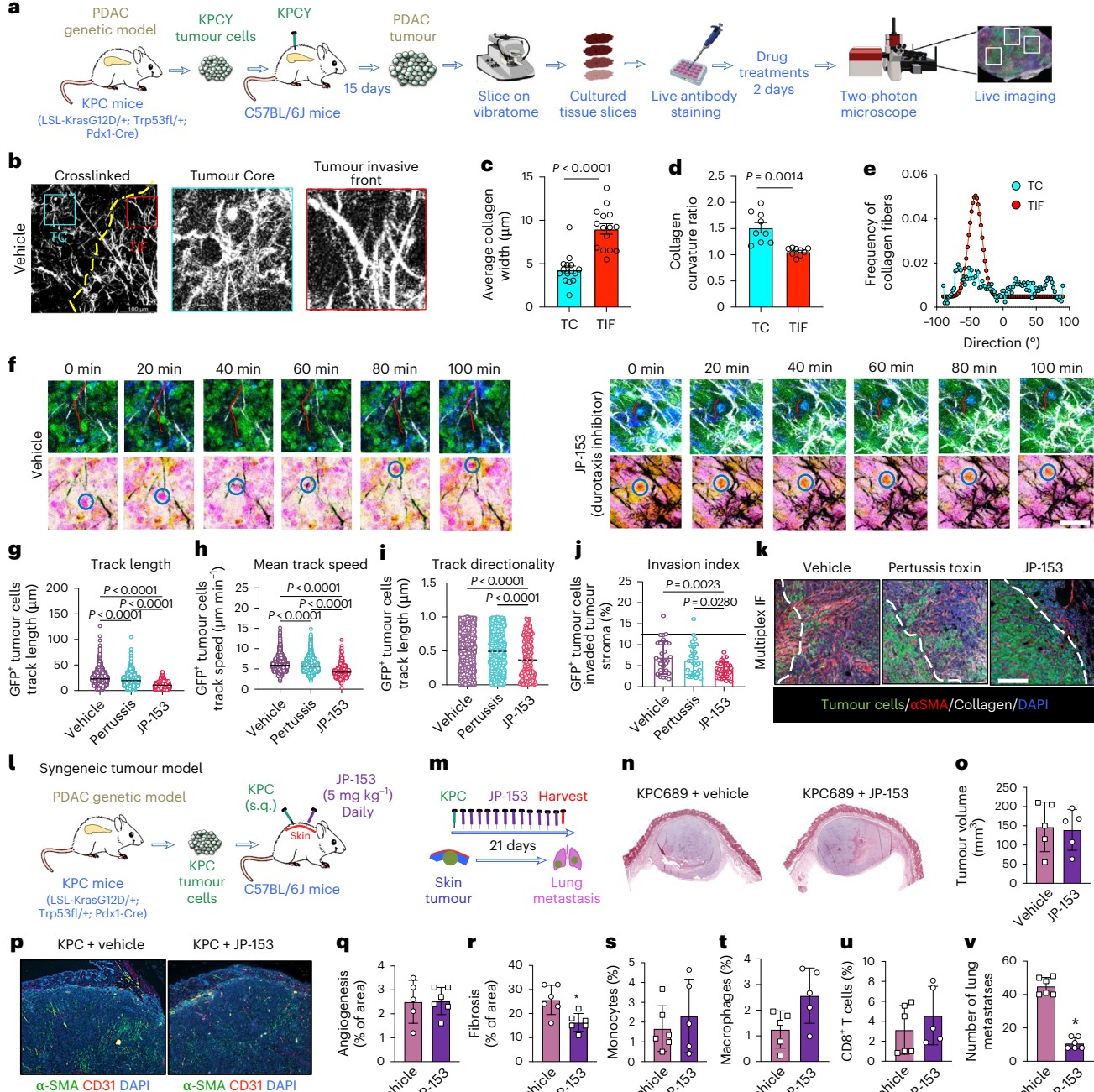

**Fig. 6 | Pharmacological inhibition of the FAK–paxillin pathway inhibits tumour cell durotaxis and metastasis in a mouse model of pancreatic cancer.** **a**, Workflow for two-photon microscopy. **b**, Second-harmonic generation (SHG) images of the collagen network at the tumour core (TC) and tumour invasive front (TIF) of tumour slices. **c**–**e**, Collagen fibre width distribution (Student's *t*-test; ****P < 0.0001 versus TC) (**c**), collagen fibre orientation distribution (Student's *t*-test; ***P < 0.001 versus TC) (**d**) and collagen fibre curvature, defined by the curvature ratio, measured from SHG images at the TC and TIF areas (**e**). **f**, Representative two-photon images at the interfaces of TC and TIF of tumour slices treated with vehicle control or JP-153 for 2 days. Migratory YFP⁺ tumour cells (green/blue in top panel and circle in bottom panel) and SHG (grey). Tracks are colour-coded according to tumour cell displacement length. Frame interval, 20 min. Scale bar, 100 μm. **g**–**j**, Quantification of YFP⁺ tumour cell motility in mean track length (**g**), mean track speed (**h**), mean track directionality (**i**) and invasion (**j**) into tumour stroma for treated groups at day 2 after administration. One-way ANOVA test. *P < 0.05, **P < 0.01, ****P < 0.0001. **k**, Representative multiplex immunofluorescence images of tumour slices after treatment with vehicle control, pertussis toxin or JP-153. Staining for tumour cells (green), α-SMA

(red), collagen (white) and DAPI (blue). **l**, A schematic of the second syngeneic tumour model involving subcutaneous injection of the pancreatic mouse cancer line (KPC689) into C57BL/6J mice. **m**, A schematic showing JP-153 (durotaxis inhibitor, 5 mg ml⁻¹, administered daily via topical microemulsion) treatment regimen in mice. **n**,**o**, The effect of JP-153 on subcutaneous flank tumour growth at 21 days, as assessed by H&E histological staining (**n**) and volume measurement (**o**). Representative experiment with *n* = 8 for each group. **p**, The effect of JP-153 on tumour fibrosis and angiogenesis in the primary tumour, assessed by immunohistochemistry. Staining for α-SMA (green), CD31 (red) and DAPI (purple). Scale bar, 100 μm. Student's *t*-test. *P < 0.05 versus vehicle. **q**,**r**, Quantification of angiogenesis (**q**) and fibrosis (**r**) in the treated groups. **s**–**u**, The effect of JP-153 on monocytes (**s**), macrophages (**t**) and CD8⁺ T cells (**u**) in the primary tumour, assessed by flow cytometry (*n* = 6 for each group). Student's *t*-test. *P < 0.05 versus vehicle. **v**, The number of lung metastases, assessed by picrosirius red staining. Scale bar, 100 μm. Student's *t*-test. *P < 0.05 versus vehicle. For cell-based assays and animal experiments, data are given as mean ± s.d. from independent experiments.

(Fig. 5e). Live two-photon imaging showed active KPCY cell migration within stiff, aligned collagen at the TIF (Fig. 6b,f). To dissect the mechanism, tumour slices were treated ex vivo for 48 h with vehicle, JP-153 (a selective FAK–paxillin durotaxis inhibitor) or pertussis toxin (a GPCR-targeting chemotaxis inhibitor). JP-153 reduced tumour cell track length (Fig. 6g), migration speed (Fig. 6h) and directionality (Fig. 6i), together lowering the invasion index (Fig. 6j). Pertussis toxin had no effect on the tumour invasion index (Fig. 6g–j), reinforcing durotaxis as the primary mechanism. Multiplex immunofluorescence further showed that KPCY cells preferentially invade along collagen-rich, CAF-dense regions at the TIF (Fig. 6k). Notably, JP-153 did not alter the orientation of collagen fibres between TC and TIF (Extended Data Fig. 7c–f) or tumour fibrosis upon 48 h of treatment ex vivo in this model (Extended Data Fig. 8a–c).

To further evaluate anti-durotactic therapy in vivo, we used a second syngeneic tumour model with KPC689 tumour cells implanted subcutaneously in WT C57BL/6J mice (Fig. 6l). JP-153 treatment (5 mg kg⁻¹) did not affect overall tumour growth or angiogenesis or induce changes in the level of basement membrane proteins, linked to tumour invasion in other models[41] (Fig. 6m–q and Extended Data Fig. 9a). However, tumour fibrosis was significantly reduced by JP-153, consistent with its anti-fibrotic effects in the lungs (Fig. 6r). This was accompanied by increased CD8⁺ cytotoxic T cells (Fig. 6s–u and Extended Data Fig. 9b–f), suggesting improved anti-tumour immunity. Finally, JP-153 significantly reduced lung metastases compared with vehicle control (Fig. 6v and Extended Data Fig. 9g), supporting a role for durotaxis in tumour dissemination. Unlike catalytic FAK inhibitors, JP-153 did not reactivate the STAT3 pathway, a known resistance mechanism[42], highlighting its therapeutic advantage (Extended Data Fig. 10),

## Discussion

Our study establishes durotaxis as a fundamental mechanism driving disease progression in vivo. By using genetic and pharmacologic tools to block durotaxis in preclinical models, we show that this mechanobiological process underlies two hallmark pathological behaviours: fibroblast recruitment and activation in fibrosis, and tumour cell dissemination in cancer. A central innovation of our work lies in the use of high-resolution AFM and intravital two-photon imaging to quantify stiffness gradients and visualize durotaxis in vivo. Our data reveal that stiffness gradients in diseased tissues are remarkably steeper than those in development and homeostasis, reaching 100–500 Pa μm⁻¹ over 20–50 μm, or 300–1,500 times steeper than gradients observed in *Xenopus* embryogenesis (~0.33 Pa μm⁻¹ over 300 μm)[43]. This magnitude appears to engage distinct mechanosensors based on the biological context, creating a therapeutic window to selectively target pathological durotaxis while preserving developmental and homeostatic processes. Supporting this, KI mice carrying a FAK^L994E mutation, disrupting the FAK–paxillin interaction, are viable with no overt phenotype, unlike global FAK or paxillin knockouts, which are embryonically lethal[33,34].

Mechanistically, we identify the FAK–paxillin axis as a durotaxis-specific sensor and provide in vivo proof of concept that targeting this pathway prevents lung fibrosis and pancreatic cancer metastasis. In fibrosis, durotaxis drives fibroblast recruitment and activation via FAK–paxillin–YAP signalling. Unlike anti-TGF-β- or anti-integrin-based therapies, often limited by on-target toxicity in epithelial and immune cells, durotaxis inhibition offers spatial and mechanistic precision. Our data show that durotaxis is selectively activated in stiff fibrotic tissue but dispensable in healthy lung, offering a safer and more targeted strategy to halt or reverse fibrosis. Our study also reveals the dynamic role of durotaxis in cancer. Tumour cells initially migrate from the soft core to the stiff invasive front, then transition beyond these peaks into softer tissue. This migration paradox suggests a mechanosensory switch, potentially driven by desensitization or

cytoskeletal reprogramming[44,45]. Current anti-metastatic therapies overlook the biophysical cues driving invasion. Our data show that mesenchymal tumour cells rely on durotaxis for early dissemination, independent of chemotactic cues. Thus, targeting durotaxis could suppress early metastatic spread, especially in desmoplastic tumours like PDAC, where stiff stroma limits drug penetration and immune access.

Future studies will require integration of advanced genetic tools with live-cell imaging to investigate durotaxis across cell types, tissues and diseases. Emerging evidence points to a broader mechanosensory landscape[46–52], including SRC kinase, MRTF, Piezo channels, TRPV1/4, non-muscle myosin IIA, microtubules, cdGAP and mitochondrial fission regulators, suggesting that durotaxis may be governed by diverse, context-specific mechanosensors. Taken together, our work establishes a foundation for next-generation genetic models that can precisely modulate durotaxis in a cell-type-specific manner, enabling a deeper understanding of stiffness-directed migration in health and disease. Finally, the development of durotaxis-targeting drugs, such as JP-153, introduces a therapeutic paradigm. By integrating genetic, biophysical and pharmacological strategies, our study paves the way for a class of 'mechano-therapeutics' to address unmet clinical needs in fibrosis and cancer.

## Online content

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

[1]Department of Medicine, Division of Pulmonary and Critical Care Medicine, Massachusetts General Hospital, Harvard Medical School, Boston, MA, USA. [2]Fibrosis Research Center, Massachusetts General Hospital, Harvard Medical School, Boston, MA, USA. [3]Center for Immunology and Inflammatory Diseases, Division of Rheumatology, Allergy and Immunology, Massachusetts General Hospital, Harvard Medical School, Boston, MA, USA. [4]Department of Molecular Pharmaceutics and Biomedical Engineering, University of Utah, Salt Lake City, UT, USA. [5]Department of Environmental Health, Harvard T. H. Chan School of Public Health, Boston, MA, USA. [6]Friedrich-Alexander-University Erlangen-Nuremberg, Erlangen, Germany. [7]Department of Biomedical Sciences, University of Pennsylvania, Philadelphia, PA, USA. [8]Department of Chemistry and Chemical Biology, Harvard University, Cambridge, MA, USA. [9]Massachusetts General Hospital Cancer Center, Harvard Medical School, Charlestown, MA, USA. [10]Department of Physiology and Biomedical Engineering, Mayo Clinic, Rochester, MN, USA. [11]Mass General Brigham Innovation, Cambridge, MA, USA. [12]Chien-Lay Department of Bioengineering, University of California, San Diego, La Jolla, CA, USA. [13]Sanford Consortium for Regenerative Medicine, La Jolla, CA, USA. [14]Department of Ophthalmology and Visual Sciences, Vanderbilt Eye Institute, Nashville, TN, USA. [15]National Center for Natural Products Research, The University of Mississippi School of Pharmacy, Oxford, MS, USA. [16]Liver Center, Division of Gastroenterology, Department of Medicine, Massachusetts General Hospital, Boston, MA, USA. [17]Boehringer Ingelheim Pharmaceuticals Inc., Ridgefield, CT, USA. [18]Department of Cancer Biology, Metastasis Research Center, UT MD Anderson Cancer Center, Houston, TX, USA. [19]Department of Medicine, Massachusetts General Hospital, Harvard Medical School, Boston, MA, USA. [20]Zenon Biotech, Boston, MA, USA. [21]These authors contributed equally: Taslim A. Al-Hilal, Maria-Anna Chrysovergi. [22]Deceased: Andrew M. Tager. ✉e-mail: david.lagares@sloan.mit.edu

## Methods

All mouse experiments were performed in accordance with National Institute of Health guidelines, and protocols were approved by the Massachusetts General Hospital Subcommittee on Research Animal Care (protocol no. 2019N000148), and all mice were maintained in a specific-pathogen-free environment certified by the American Association for Accreditation of Laboratory Animal Care. In animal experiments, data distribution was assumed to be normal, but this was not formally tested. Data collection and analysis were not performed blind to the conditions of the experiments.

All human experiments were performed under protocols approved by the Institutional Ethics Committee approved by the Massachusetts General Hospital. PDAC cell lines were derived and established from metastatic ascites collected under a discarded tissue protocol, in full compliance with Massachusetts General Hospital IRB guidelines (protocol no. 2011P001236), as previously reported[36]. These tumour cell lines were generated years ago; therefore, no identifying patient information or compensation records are available to the corresponding author. These tumour cell lines were generously provided by D.T. and M.L. (MGH Cancer Center and Harvard Medical School). Requests for additional patient-related information should be directed to these authors.

### Animals

Pathogen-free male C57BL/6N (6–8-week-old) mice purchased from the National Cancer Institute Frederick Mouse Repository were used for mouse models of skin, lung and kidney fibrosis as well syngeneic tumour models. Immunocompromised NOD/SCID/gamma-c (NSG; NOD.Cg-Prkdc$^{scid}$ Il2rg$^{tm1Wjl}$/Sz, 6–8–week-old) mice obtained from The Jackson Laboratory were used for the orthotopic xenograft model.

### Mouse model of lung fibrosis

Lung fibrosis in 6–8-week-old mice was induced by intratracheal administration of bleomycin (50 μl at 1.2 U kg$^{-1}$ body weight), as previously described[19,21,53]. Sterile saline was used as control. Mice were euthanized at the indicated timepoints including day 3 (peak of epithelial cell injury), day 7 (peak of inflammation) and day 21 (peak of fibrosis). Lungs and BAL were collected for AFM, histologic, flow cytometry, molecular and biochemical studies as well hydroxyproline analysis.

### Mouse model of skin fibrosis

Skin fibrosis in 6–8-week-old mice was induced by daily subcutaneous injection of bleomycin (100 μl from 10 μg ml$^{-1}$ stock) for 28 days, as previously described[21]. Sterile saline was used as the control. At the conclusion of experiments, mice were euthanized, and full-thickness 6-mm punch biopsies were obtained for AFM, histologic, molecular and biochemical studies as well as hydroxyproline analysis.

### Mouse model of kidney fibrosis

Renal fibrosis in 6–8-week-old mice was induced by unilateral ureteral obstruction (UUO) surgery, as previously described. In brief, the left ureter was ligated using a suture, leading to an obstruction of the kidney outflow tract on the ligated side so that the urine cannot drain anymore, causing hydronephrosis with tubular dilation and kidney fibrosis at day 14. Kidneys were collected for AFM, histologic, molecular and biochemical studies as well as hydroxyproline analysis.

### Orthotopic xenograft mouse model of human pancreatic cancer

Pancreatic fibrotic tumours were induced by co-injecting $1 \times 10^5$ PDAC3 GFP-Luciferase-tagged cancer cells with $9 \times 10^5$ CAF-1 cells (PDAC-3%:CAF-1%, 1:9 ratio) in Matrigel (50 μl) and Dulbecco's modified Eagle medium (50 μl, 1:1. ratio) into the tail of the pancreas, as previously described[36]. Pancreatic tumours were monitored every 3 days using in vivo luciferase imaging. In brief, mice were injected intraperitoneally with 100 mg kg$^{-1}$ of body weight of luciferin

(200 μl of a 10 mg ml$^{-1}$ solution of luciferin in phosphate-buffered saline (PBS)) 10–15 min before imaging and anaesthetized with isoflurane, and bioluminescent signal (photon flux) was measured from the abdominal region of interest using the In Vivo Imaging System (IVIS) Lumina platform (PerkinElmer, Caliper).

### Syngeneic KPC tumour model

Dermal fibrotic tumours were induced by co-injecting $5 \times 10^5$ cells in 100 μl PBS of the metastatic KPC689 pancreatic tumour line isolated from KPC mice (Pdx1-Cre; LSL-Kras$^{G12D/+}$; LSL-Trp53$^{R172H/+}$), as previously described[54].

### Collagen-GFP reporter mice

To facilitate the study of fibroblast biology in vivo, we used a transgenic reporter mouse expressing GFP under the control of collagen type I promoter (col-GFP mice) (The Jackson Laboratory, (Col1a1*2.3-GFP)1Rowe/J)). These mice are on a C57BL/6J background, develop robust lung fibrosis after bleomycin challenge and efficiently label collagen-producing cells with high specificity.

### Generation of FAK$^{L994E}$ KI mice

FAK$^{L994E}$ KI mice were developed at Taconic Biosciences via CRISPR–Cas9-mediated gene editing in C57BL/6 mice. In brief, the L994E mutation was introduced into exon 32 of the mouse Ptk2 gene (FAK, NCBI Gene ID: 14083; Ensembl Gene ID: ENSMUSG00000022607) via CRISPR–Cas9 with specific guide RNA (gRNA) and a short single-stranded DNA template. The gRNA sequence targeting exon 32 is GCTGAATCCGCTCGAGTAGT. The TTA codon (AAT complementary sequence in gRNA) indicates leucine 994, which was targeted for mutation with a short single-stranded DNA template including the CTT codon sequence, thus replacing leucine 994 by glutamic acid (L994E). An additional silent mutation was inserted into exon 32 to generate a restricition site (Hphl) for analytical purposes. In vitro fertilization was performed using oocytes from superovulated C57BL/6NTac females. For microinjection, one-cell-stage embryos were injected with a Cas9–gRNA ribonucleoprotein complex and single-stranded oligodeoxynucleotides (ssODNs) into the pronucleus of each embryo. After recovery, 25–35 injected one-cell-stage embryos were transferred to one of the oviducts of 0.5 days post coitum (dpc), pseudopregnant Naval Medical Research Institute (NMRI) females. Genotyping analysis was performed by polymerase chain reaction (PCR) using genomic DNA extracted from biopsies. The following templates were used as controls: H$_2$O (ctrl1) and WT genomic DNA (ctrl2). The PCR amplicons were analysed using a Caliper LabChip GX device. The PCR detects the CRISPR–Cas9-induced constitutive KI allele as well as potential indel modifications and the unmodified WT allele. To distinguish indel modifications from unmodified WT sequences, a heteroduplex analysis (for example, via capillary electrophoresis) was performed. HphI digest results in cleavage of the 380-bp PCR product in two fragments (257 bp and 123 bp). Primers: forward primer: TCTTGGTGGCTCAAAGACAG; reverse primer: GGGCTACAGAGGCTAAGGTTAC. Expected fragments (bp): 380 (wt), 380 (indel), 380 (HDR). Furthermore, a restriction analysis of PCR product was perform to validate the presence or absence of the intended mutation via homology-directed repair. See the Supplementary Methods for further validation of this mouse model.

### Generation of mouse precision-cut tumour slices and two-photon imaging

Tumours were embedded in a solution of 5% low-gelling temperature agarose (Sigma-Aldrich) in PBS. Subsequently, the tumours were sliced to a thickness of 500 μm using a Leica VT1200S vibratome immersed in ice-cold PBS. After slicing, live tumour slices were treated with AF647-anti-mouse CD90.2 (obtained from BioLegend) at a concentration of 10 μg ml$^{-1}$ for 15 min at 37 °C. These slices were then transferred to 0.4-mm organotypic culture inserts (from Millipore)

within 35-mm Petri dishes containing 1 ml RPMI-1640 (no phenol red; ThermoFisher) for subsequent treatment and imaging. Imaging of the fresh mouse tumour slices was conducted using a Leica SP8-MP upright multiphoton microscope with a Coherent Chameleon Vision II MP laser, equipped with a 37 °C thermostatic chamber. Tumour slices were secured using a stainless-steel ring slice anchor (Warner Instruments) and were perfused with RPMI-1640 solution (no phenol red) bubbled with 95% $O_2$ and 5% $CO_2$ at a rate of 0.3 ml min$^{-1}$. Imaging was systematically performed at six different regions within the tumour using a 20× (1.0 numerical aperture) water immersion lens and a Coherent Chameleon laser at 880 nm/25 mW. Fluorescence detection utilized CFP (483/32), GFP (535/30), AF647 (685/40), and tdTomato (610/75) filters.

For the analysis of cell migration in four dimensions, $z$ stacks of 70–90 μm with a step size of 5 μm were acquired every 30 s for 2 h, alternating between six fields. Videos were generated by compressing the $z$-stack information into a single plane using the max intensity $z$ projection feature of Imaris and LAS X software. Cellular motility parameters were then calculated using Imaris software, with tracks covering more than 10% of the total recording time included in the analysis. Any drift in the $x$, $y$ dimensions was corrected using the 'Correct 3D Drift' plug-in in FIJI-ImageJ. Quantification of tumour cell number and motility in various tumour regions, including stroma-rich and tumour invasive regions, was performed. These regions were delineated on the basis of visual inspection of immunofluorescence images. Fluorescence intensities were determined in regions of interest using FIJI-ImageJ, and the number of migratory tumour cells in defined regions was quantified using the Analyze Particles function of FIJI-ImageJ after thresholding and conversion to binary images. The directionality of cell movement was calculated by evaluating the ratio of the shortest distance between the start and end points of a cell's trajectory to the total distance it travelled. This metric provides a measure of how directly the cell moved towards its destination, with values ranging from 0 to 1, where a straightness value close to 1 indicates highly linear movement (straight path) and a value closer to 0 suggests a more meandering or random movement pattern. Collagen measurement was conducted using CT-FIRE software (version 2.0 beta).

### JP-153 administration to mice
JP-153 was synthesized in the laboratory of Dr Rates. JP-153 (5 mg kg$^{-1}$) was loaded into a topical microemulsion for in vivo administration. In brief, a microemulsion system was prepared daily by dissolving 2 mg of JP-153 in 10 μl dimethyl sulfoxide (DMSO; #D5879, Sigma) from which 7.5 μl of the JP-153 stock was mixed with 63 μl of Capriol 90 (#3254, Gattefossé), 63 μl of triacetin (#W200700, Sigma) (10.5% final volume concentration) and 147 μl of Tween-20. The mixture was brought to volume (600 μl) by dropwise addition of dH$_2$O (~172 μl) with frequent gentle mixing by vortex. Before JP-153 topical application, mice were shaved behind the neck (0.5 cm$^2$). Mice were dosed under mild isoflurane anaesthesia followed by the application of 50 μl of formulated JP-153 to the shaved area.

### Assessment of fibrosis by histological analyses
The extent of fibrosis was assessed by haematoxylin and eosin (H&E), picrosirius red and Masson's trichrome staining, according to our standard protocols[21].

### Hydroxyproline assay
Collagen content in mouse tissues was measured using hydroxyproline assay, according to our standard protocols[21].

### Mouse BAL recovery
To obtain BAL samples, mouse lungs were lavaged with six 0.5-ml aliquots of PBS. BAL samples were centrifuged at 3,000$g$ for 20 min at 4 °C and transferred the supernatants to siliconized low-binding Eppendorf tubes (PGC Scientifics) for subsequent analysis.

### Vascular leak assay
Total protein concentration in BAL samples was determined using a commercially available bicinchoninic acid (BCA) Protein Assay Kit (Pierce) per the manufacturer's protocol.

### Determination of mouse TGF-β1 levels in BAL
TGF-β1 levels in mouse BAL fluids were determined by ELISA (R&D Systems, cat. no. DB100B) according to the manufacturer's protocol.

### Multiplex flow cytometry
Single-cell suspensions were isolated from mouse lung tissues biopsies using Liberase Blendzyme (final concentration, 0.14 U ml$^{-1}$; Roche) and deoxyribonuclease I (final concentration, 60 mg ml$^{-1}$; Sigma) for 45 min at 37 °C. Cells were incubated with FcRII and FcRIII blocking antibody (BioLegend, clone 93) for 10 min at 4 °C followed by staining with the following fluorophore-conjugated antibody from BioLegend: Viability eF780 (1:1,000), CD11b-BUV737 (1:100), Ly6G-FITC (1:200), Ly6C-PerCP-Cy5.5 (1:200), CCR2-PE (1:50), CD11c-BV605 (1:200), MHCII-Pe-Cy7 (1:1,000), F4/80-PE (1:100), MerTK-APC (1:100), CD3-BUV395 (1:200), CD4-BV786 (1:200), CD8-FITC (1:200). Flow cytometry was performed using a BDLSRFortessa X-20 cell analyser, and FlowJo software was used for analysis.

### Tumour burden and metastatic index
IVIS imaging of tumours in the pancreas, skin, livers and lungs was performed immediately after euthanizing the mice. Normalized metastatic tumour burden (metastatic index) was calculated by dividing the total amount of photon flux from liver and lungs of each animal by the photon flux of its primary tumour (pancreas or skin). Exposure conditions (time, aperture, stage position and binning) were kept identical for all measurements within each experiment, as previously described[36].

### Atomic force microscopy
AFM was performed according to our standard techniques using Bio-Catalyst AFM (Bruker) and MFP-3D AFM (Asylum Research)[21,30]. In brief, force–indentation profiles were acquired from thin mouse lung, skin, kidney and pancreas tissue slices by performing microindentations at points separated by 5 μm spatially covering an 80 × 80 μm area. An sphere-tipped probe (Novascan) with a diameter of 5 μm and a nominal spring constant of ~60 pN nm$^{-1}$ was used. The cantilever spring constant was further confirmed by the thermal fluctuation method. Force curves were performed, and indentation profiles were acquired at an indentation rate of 20 mm s$^{-1}$ for a force of around 16 nN applied on the tissue. Elastic modulus (Young's modulus) was estimated by fitting force curves with the Hertz contact model (Hertz 1881; Dimitriadis 2022) following $E = 3/4((1 - v^2))/(R^{(1/2)} \delta^{(3/2)})F$, where $R$ is the tip radius, $\delta$ is the sample indentation and $v$ is the Poisson's ratio assumed at 0.4 for tissue. Resulting young modulus data were plotted in 3D stiffness maps using MATLAB.

### Co-registration of fibroblasts and myofibroblasts with matrix stiffness
Areas of active fibrosis were identified by accumulation of GFP$^+$ collagen-producing cells by fluorescence microscopy coupled to AFM mapping. To differentiate between fibroblasts and myofibroblasts, tissues were fixed after AFM mapping and stained with antibodies against GFP and α-SMA. Fibroblasts were identified as GFP$^+$/α-SMA$^-$ positive cells, whereas myofibroblasts were identified as GFP$^+$/α-SMA$^+$ double-positive cells. Post-analysis focused on fibroblasts versus myofibroblast-matrix stiffness co-localization.

### Fabrication of PA hydrogels with steep stiffness gradients
Photolithography was used to bioengineer PA hydrogels with alternating stiff and soft stripes ('zebraxis'), containing step stiffness gradients between them, as previously described[55]. First, master

Si wafers (SU-8 2015) were patterned with 25-mm-long by 100-µm-wide by 20-µm-high cuboids spaced 200 µm apart using soft photolithography. Our photomask was designed in AutoCad (CAD/Art Services). A soft PA hydrogel (4 kPa, 20–22 µL) was initially photo-polymerized on top of a methacrylate-treated 18-mm coverslip using ultraviolet (UV) light (365 nm) for 5 min. IrgaCure (5%) was used as a photo-initiator. The soft gel was dehydrated for 1 h at 30 °C. A second PA solution (40 kPa, stiff hydrogel, 20–22 µl) was added on top of the soft hydrogel and photo-crosslinked for 5 min in the form of stripes by using our photomask placed between the UV light source and the coverslip with the hydrogels. This photomask limits the light to penetrate through the stiff hydrogel by blocking the light on a striped fashion. Ultimately, the photomask generates with crosslinked stiff stripes (100 µm) and unpolymerized stripes (200 µm). Lastly, the unpolymerized stripes were washed with PBS. The end product results in an intercalating stripes of soft and stiff hydrogels (zebraxis hydrogel). Many hydrogels were fabricated simultaneously from the same polymer solutions to limit batch-to-batch variability. Stiffness hydrogels were then activated with Sulfo-SANPAH (1 mg ml$^{-1}$) and functionalized with fibronectin at 10 µg ml$^{-1}$ (Sigma) on PBS for 4 h at 37 °C. Two hydrogels per batch were mechanically characterized by AFM and coating efficiency was measured by immunofluorescence, using our standard protocols. Designing another photomask, we also engineered an island of soft matrix (1 kPa) surrounded by alternating soft (1 kPa) and stiff (25 kPa) stripes to investigate durotaxis capacity of pancreatic cancer cells. The stiffness of these 'gradient' gels increases from 1 kPa to 25 kPa (mimicking the range of stiffness produced in vivo).

### Durotaxis assay on steep stiffness gradients

Cells were plated with a density of 10,000 cells per 20-mm hydrogel in serum-free medium on zebraxis hydrogels. Cells were equally distributed at 50% over soft versus stiff stripes at 4 h after plating (durotactic index of 1). Durotaxis was measured at 24 h by calculating the ratio of cells over stiff/soft stripes upon fixation and staining with 4′,6-diamidino-2-phenylindole (DAPI). Five images at 10× resolution were counted per hydrogel. Experiments were performed in triplicate.

### Fabrication of hydrogels with shallow stiffness gradients using microfluidics

Microfluidic fabrication was used to bioengineer PA hydrogels containing shallow stiffness gradients, as previously described[22]. In brief, the microfluidic channels were filled with polymer solutions consisting of 10% acrylamide and either 0.05% (low) or 0.5% (high) bis-acrylamide in deionized water using a syringe pump (KD Scientific) at 30 µl min$^{-1}$ into the three inlets in this order: low–high–low. After the solutions split and recombined, the polymerization was initiated by turning on the UV transilluminator for 6 min located directly beneath the outlet portion of the microchannel.

### Analysis of durotaxis by time lapse

Cells were plated with a density of 15,000 cell hydrogel in serum-free medium and imaged using time-lapse microscopy. Cells were imaged for 24 h at 5-min intervals with a 10× 0.3 numerical aperture Plan Fluor objective lens. Trajectories of 75 cells in the observation area of each image sequence were tracked manually using the National Institutes of Health (NIH) ImageJ Manual Tracking plug-in. Several values characterizing cell migration were computed from the trajectories by the software, such as forward migration indices, which express the directionality of migration. Experiments were performed in triplicate.

### PA hydrogels

PA hydrogels were prepared as described previously[21]. In brief, 18-mm glass coverslips (Fisher Scientific) were treated with a 0.4% solution of 3-methacryloxypropyltrimethoxysilane (Sigma-Aldrich) in acetone for 20 min, rinsed once with fresh acetone and air dried. Solutions of variable ratios of acrylamide:bis-acrylamide (Bio-Rad) were prepared to fabricate hydrogels of 0.5 kPa (3:0.11 ratio) and 64 kPa (20:0.24 ratio). The hydrogels were functionalized by incubation for 30 min with 0.05 mg ml$^{-1}$ sterile collagen type I (PureCol, Advanced BioMatrix) in PBS.

### Generation of paxillin lentiviruses

Non-replicative lentiviruses were produced by transient transfection of human embryonic kidney (HEK) 293T cells of paxillin constructs in combination with regulator of expression of virion proteins (REV), vesicular stomatitis virus glycoprotein (VSVG) and promyelocytic leukemia protein (PDML) (Addgene) using Lipofectamine 2000 (Invitrogen) with plasmids. Viral supernatants were collected 48 h after transfection and concentrated with Lenti-X Concentrator (Clontech). Cells were transduced with lentivirus for 6 h in 6 mg ml$^{-1}$ polybrene, according to our standard laboratory protocols[21].

### Cell treatments

JP-153 was first solubilized in DMSO and then diluted into serum-free culture medium on a final stock concentration of 10 mM (<0.01% (v/v) DMSO). The stock concentration was aliquoted and kept at −20 °C until further use. Cells were treated with different concentrations of JP-153 from 10 nM up to 10 µM on serum-free medium for 24 h before performing cell-based assays including chemotaxis, durotaxis, invasion and crystal violet. To assess myofibroblast activation induced by TGF-β1, cells were treated with JP-153 for 24 h and with TGF-β1 (5 ng ml$^{-1}$) + JP-153 (100 nM) for another 24 h.

### Proximity ligation assay

FAK–paxillin interactions were detected via PLA using the Duolink In Situ Red Starter Kit Mouse/Rabbit (Sigma), following the kit protocol. Primary antibody incubations were performed using a rabbit polyclonal FAK antibody, 1:100 (#3285S, Cell Signaling), and a mouse polyclonal paxillin antibody, 1:100 (#AF4259, R&D Systems). For quantification analysis, ImageJ software (NIH) was used.

### Statistics and reproducibility

Sample sizes were calculated by power analysis for fibrosis and cancer studies. In each experiment evaluating the effects of inhibition of FAK–paxillin by JP-153 on the extent of lung fibrosis produced in mice, $n \geq 8$ mice per group were used to achieve statistical significance. Sample sizes were estimated on the basis of 80% power to detect a 50% reduction in the amount of fibrosis present in mice treated with JP-153 compared with WT control mice, accepting a type I error rate of 0.05. Animals were distributed into groups of equal body weight. No animals were excluded from the analysis. In each experiment evaluating the effects of inhibition of FAK–paxillin by JP-153 on the tumours and metastases produced in mice, we used a power calculation approach assuming we wanted to detect a minimum difference between groups of twofold in tumour growth at 3 weeks from initial injection. Assuming a coefficient of variation less than 45%, we decided to have a minimum of six mice per arm, which allowed us to detect twofold differences with a power of 80%, accepting a type I error rate of 0.05. Experimental data were analysed by unpaired Student's $t$-test for differences between each of the experimental conditions, two-way analysis of variance (ANOVA) for overall condition effects, two-tailed Spearman rank for correlation, or nonlinear dose–response curve fitting for half-maximal effective concentration (EC$_{50}$) calculation using GraphPad Prism 5.0 software. The $P$ values obtained are indicated in the figures and figure legends when statistically significant. $P < 0.05$ was considered significantly different between groups. All data displayed a normal distribution. Data are reported as mean ± s.d.

**Reporting summary**

Further information on research design is available in the Nature Portfolio Reporting Summary linked to this article.

## Data availability

All data supporting the findings of this study are available from the corresponding author on reasonable request. Source data are provided with this paper.

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

## Acknowledgements

D.L. acknowledges funding support from the NIH (grant numbers R01 HL157384-01A1 and R01 HL147059-01). D.J.T. acknowledges funding support from from the NIH (grant number R01 HL092961).

## Author contributions

D.L. and A.M.T. conceived the idea, designed and supervised all experiments and contributed to manuscript preparation. T.A.A.-H., M.-A.C., P.E.G., F.L., V.A., Y.Z., M.A.L.D., A.S., T.I., C.K.P., V.V., T.S.R., W.H.G. and J.L.A. performed the experiments, analysed the data and contributed to manuscript preparation. Z.X. performed the experiments with precision-cut tumour slices under the supervision of E.P. E.C. and C.R.Y. provided JP-153, and C.M.W. characterized JP-153. T.A.A.-H., L.V., C.H. and A.J.E. helped in developing the bioengineering models. M.L., D.T. and R.K. provided the cell lines and helped in developing the PDAC models. K.D., A.C.M. and E.S.W. helped in working with the fibrosis model. D.S. performed Bio-AFM studies under the supervision of D.J.T. T.A.A.-H. and D.L. wrote the manuscript, and E.P. and D.J.T. reviewed and edited the manuscript.

## Competing interests

D.L. is a founder and has a financial interest in both Mediar Therapeutics and Zenon Biotech. The companies are developing treatments for organ fibrosis and cancer related to this work. D.L. has received consulting fees from Merck & Co, Scholar Rock, Ono Pharma, UCB Biopharma, Calico Life Sciences, Johnson & Johnson, Inzen Therapeutics, BioHope and PureTech Health LLC that are not related to this work. D.L. has received research support from Boehringer Ingelheim, Merck & Co, Indalo Therapeutics, Ono Pharma and Unity Biotechnology, which was not used in this work. D.J.T. is a scientific advisor and a financial interest in Zenon Biotech. D.T. has received consulting fees from ROME Therapeutics, Foundation Medicine, Inc., NanoString Technologies, EMD Millipore Sigma and Pfizer that are not related to this work. D.T. is a founder and has equity in ROME Therapeutics, PanTher Therapeutics and TellBio, Inc., which is not related to this work. D.T. receives research support from ACD-Biotechne, PureTech Health LLC and Ribon Therapeutics, which was not used in this work. D.L.'s and D.T.'s interests were reviewed and are managed by Mass General Brigham in accordance with their conflict-of-interest policies. The other authors declare no competing interests.

## Additional information

**Extended data** is available for this paper at https://doi.org/10.1038/s41556-025-01697-8.

**Correspondence and requests for materials** should be addressed to David Lagares.

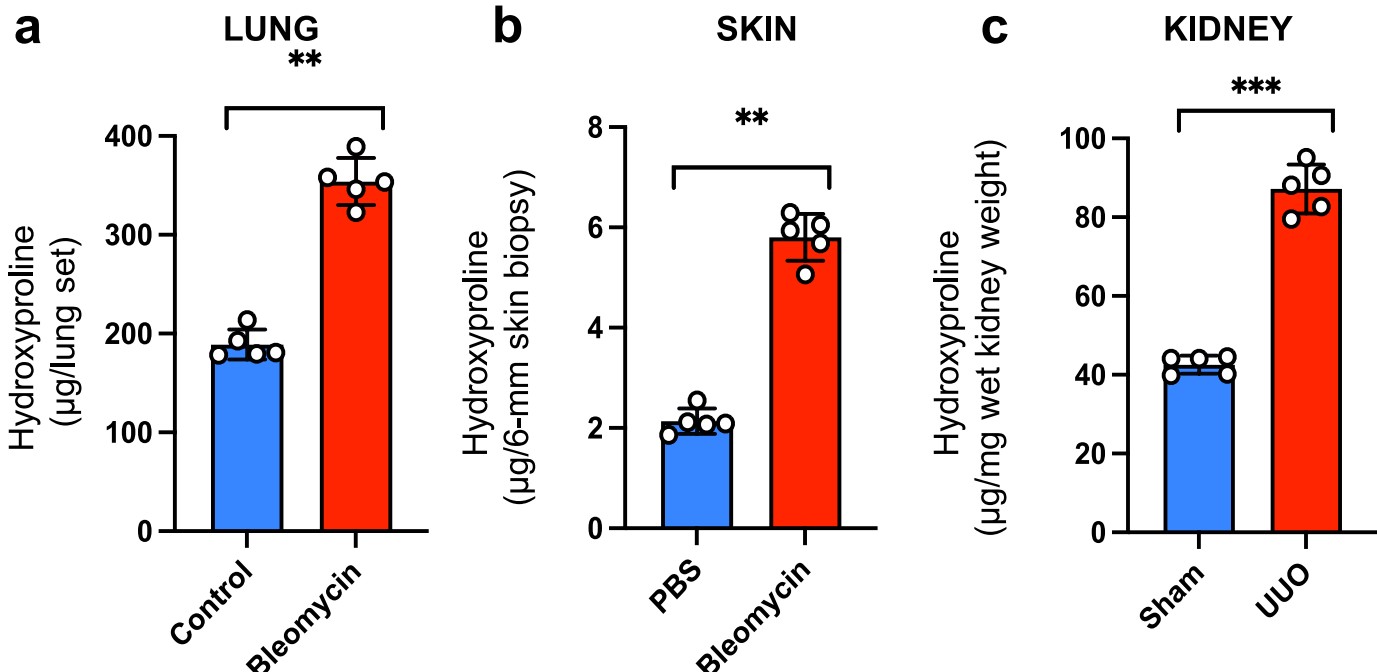

**Extended Data Fig. 1 | Increased collagen deposition in fibrotic tissues from mouse models of skin, lung, and kidney fibrosis. (a,b,c)** Hydroxyproline content (biochemical marker of collagen deposition) measured in the lungs, skin, and kidneys of mice subjected to the mouse models described in Fig. 1. Data are given as mean ± SD with n = 6 for each group. P value was determined by Student's t test.

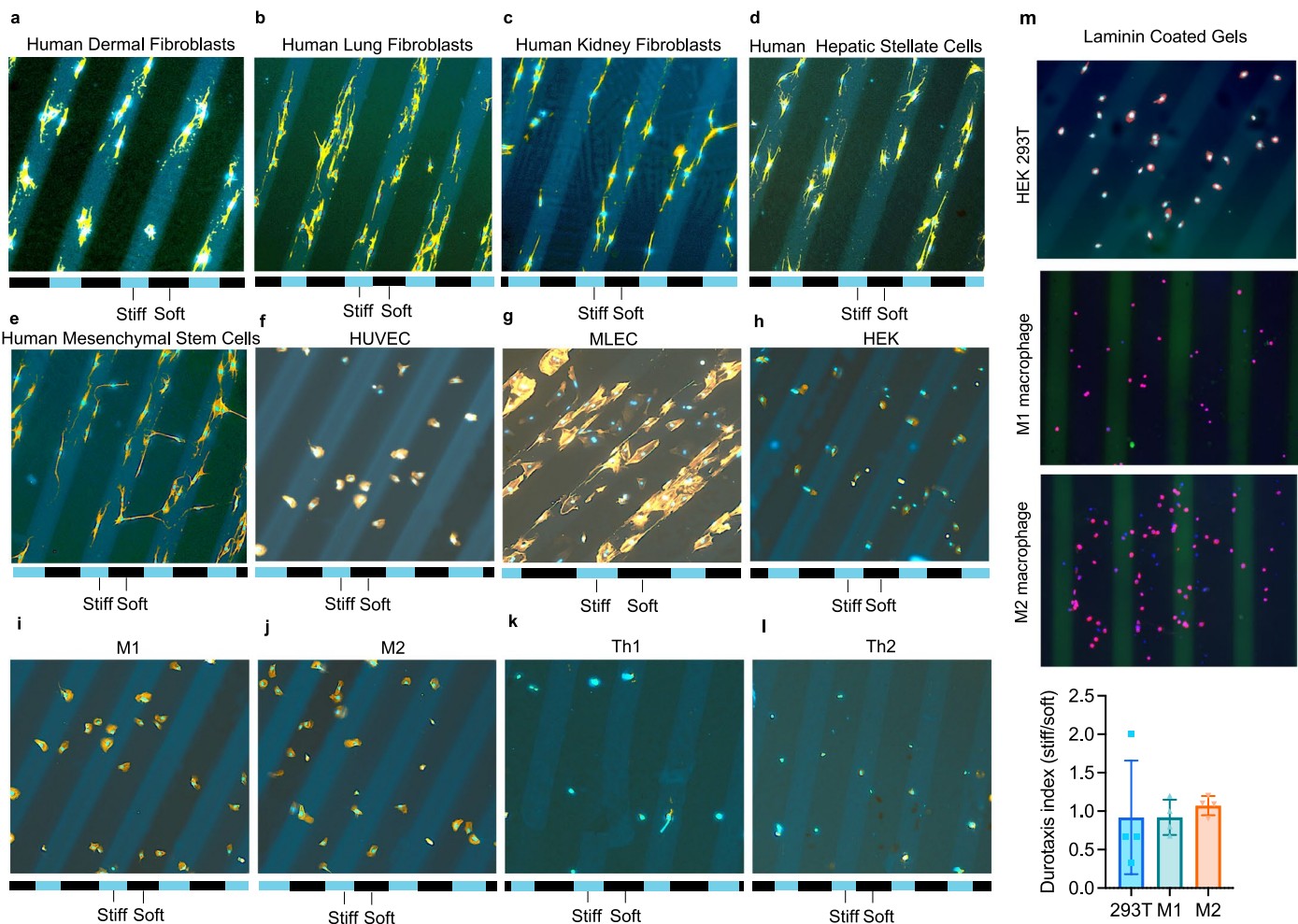

**Extended Data Fig. 2 | Assessment of durotaxis in multiple human and mouse cells involved in tissue fibrosis on mechanically patterned hydrogels.** Durotaxis capacity was assessed on human fibroblasts isolated from skin (**a**), lungs (**b**), and kidneys (**c**), human hepatic stellate cells (**d**), human mesenchymal stem cells (**e**), human umbilical vein endothelial cells (HUVECs, **f**), mouse lung endothelial cells (MLEC, **g**), human embryonic kidney 293 cells (HEK 293, **h**), mouse pro-inflammatory M1 (**i**) and pro-fibrotic M2 polarized macrophages (**j,k**) and mouse T helper type 1 (Th1) and Th2 cells (**k,l**). Durotaxis was determined by quantifying the ratio of the number of cells accumulated on stiff versus soft stripes at 24 h after plating. Cells were identified by staining with phalloidin (red) to visualize F-actin and 4′,6-diamidino-2-phenylindole (DAPI; blue) to visualize nuclei. Representative images from 3 biological replicates. (**m**) Durotaxis capacity was assessed on epithelial and macrophages using laminin-coated substrates.

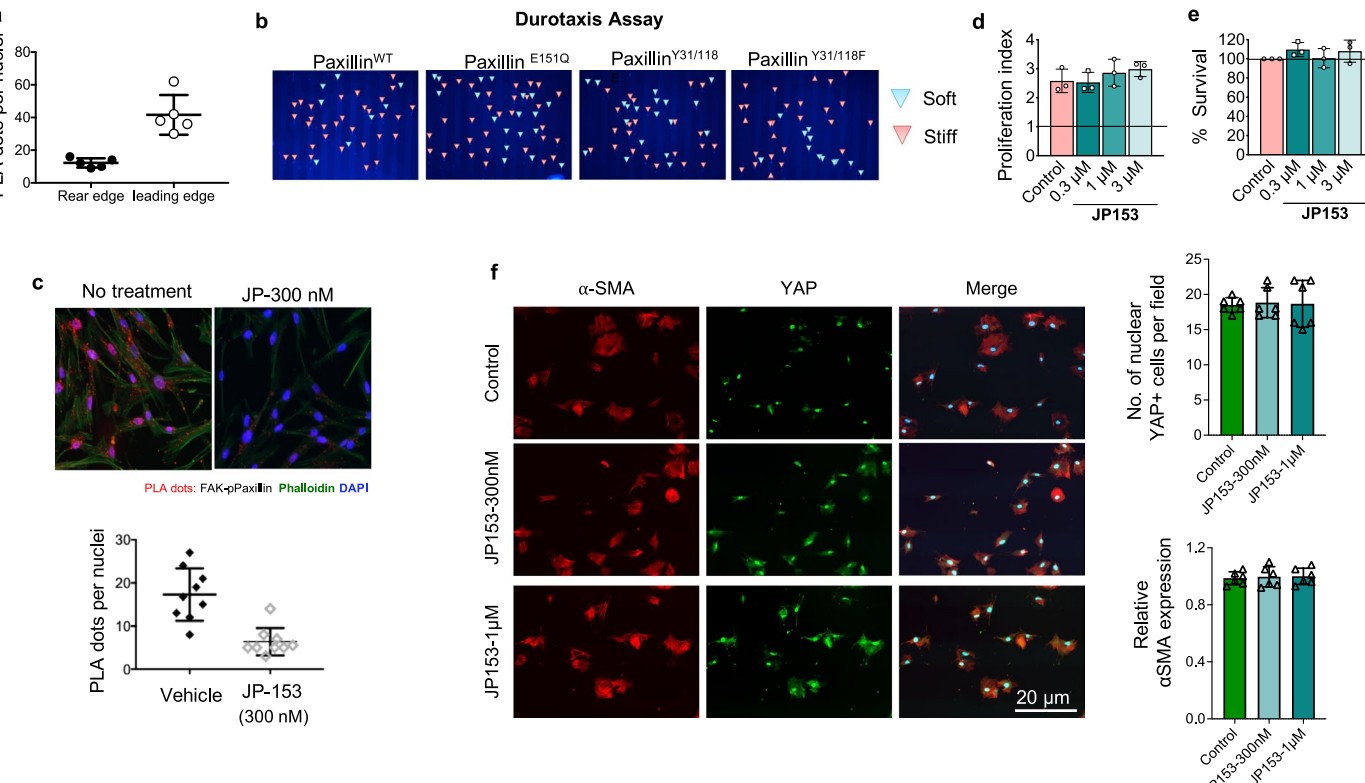

**Extended Data Fig. 3 | Genetic or pharmacological inhibition of the FAK-Paxillin^{Y31/118} pathway inhibits fibroblast durotaxis *in vitro*.**
(**a**) Activation of the FAK-Paxillin^{Y31/118} pathway at the leading edge of durotaxing fibroblasts, assessed by proximity ligation assay (PLA). Quantification from data shown in Fig. 3c. Data are given as mean ± SD with n = 5 for each group. P value was determined by Student's t test. (**b**) Effect of lentiviral overexpression of paxillin mutants including constitutively active phosphomimetic paxillin (paxillin^{Y31/118E}), inactive non-phosphorylatable paxillin (paxillin^{Y31/118F}), or paxillin point mutant defective in vinculin binding (paxillin^{E151Q}) on fibroblast durotaxis on mechanically patterned hydrogels. Red arrows indicate cells on stiff bars, whereas blue arrows mark cells on soft stripes. Representative images from

3 biological replicates. (**c**) Effect of JP-153 (300 nM) on FAK-Paxillin interaction in lung fibroblasts, assessed by PLA. Cells were identified by staining with phalloidin (green) to visualize F-actin and 4′,6-diamidino-2-phenylindole (DAPI; blue) to visualize nuclei. Red dots indicate FAK/phosphoPaxillin Y31 interaction. Scale bar 10 μm. Representative images from 8 biological replicates. (**d,e**) Dose response effect of JP-153 on fibroblast proliferation and survival at 48 h. (**f**) JP-153 has no effect on α-SMA expression or YAP cytoplasmic to nuclear translocation on stiffness-activated myofibroblasts. Fibroblasts were cultured on stiff-hydrogels (25 kPa) for 3 days and treated with or with JP-153 at 1 micromolar and 300 nM for addition 24 h. n = 3. Data are given as mean ± SD with n = 3 for each group. P value was determined by Student's t test.

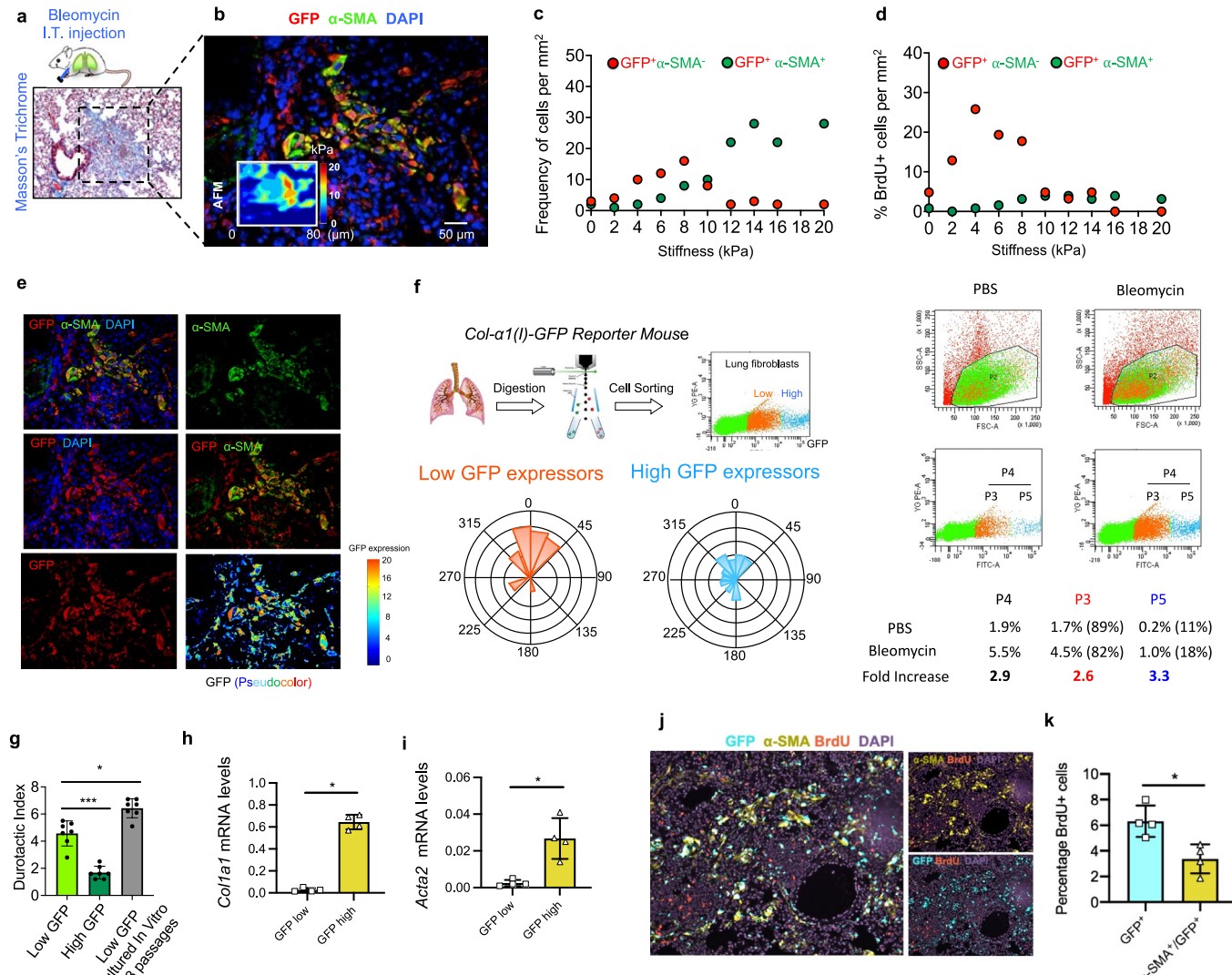

**Extended Data Fig. 4 | Assessment of durotaxis in different fibroblast populations isolated from the bleomycin-induced lung fibrosis model.**
(**a-e**) Co-registration of matrix stiffness by atomic force microscopy (AFM) and fibroblast position by immunofluorescence shows that GFP+α-SMA+ fibroblasts locate at areas of high stiffness, while GFP+α-SMA- fibroblasts reside in regions of lower stiffness. Pseudocolor imaging revealed low GFP expression levels (GFPlow) in lung fibroblasts compared to high GFP expression levels (GFPhigh) observed in α-SMA+ myofibroblasts. One representative experiment is presented from n = 4. Scale bar 20 µm. (**f**) Percentage of GFPhigh and GFPlow fibroblasts in healthy and fibrotic lungs. Representative experiment from n = 4 per group. (**g**) Assessment of durotactic capacity of GFPlow/α-SMA- fibroblasts and GFPhigh/α-SMA+ myofibroblasts isolated from fibrotic lungs via fluorescence-activated cell sorting (FACS), and GFPlow fibroblasts after passaging three

times. These experiments used hydrogels fabricated by microfluidic gradient generator and cells were allowed to durotax for 7 days. Results showed that GFPlow fibroblasts and GFPhigh fibroblasts demonstrate very distinct durotactic capacities, while GFPlow fibroblasts durotax up stiffness gradients with great directionally, GFPhigh fibroblasts barely durotax in this in vitro system. (**h,i**) Assessment of collagen type I (Col1a1) and α-SMA (Acta2) mRNA levels by real time PCR in GFPhigh and GFPlow fibroblasts isolated from healthy and fibrotic lungs via fluorescence-activated cell sorting (FACS). (**j,k**) Assessment of fibroblast and myofibroblast proliferation in vivo by BrdU staining in Col-GFP mice at 21 days post-bleomycin challenge. Paraffin-embedded lung sections were stained for BrdU (orange), GFP (blue), α-SMA (yellow), and DAPI (purple) to visualize proliferating GFPhigh/α-SMA+ myofibroblasts and GFPlow/α-SMA- fibroblasts. One representative experiment from n = 5. Scale bar 100 µm.

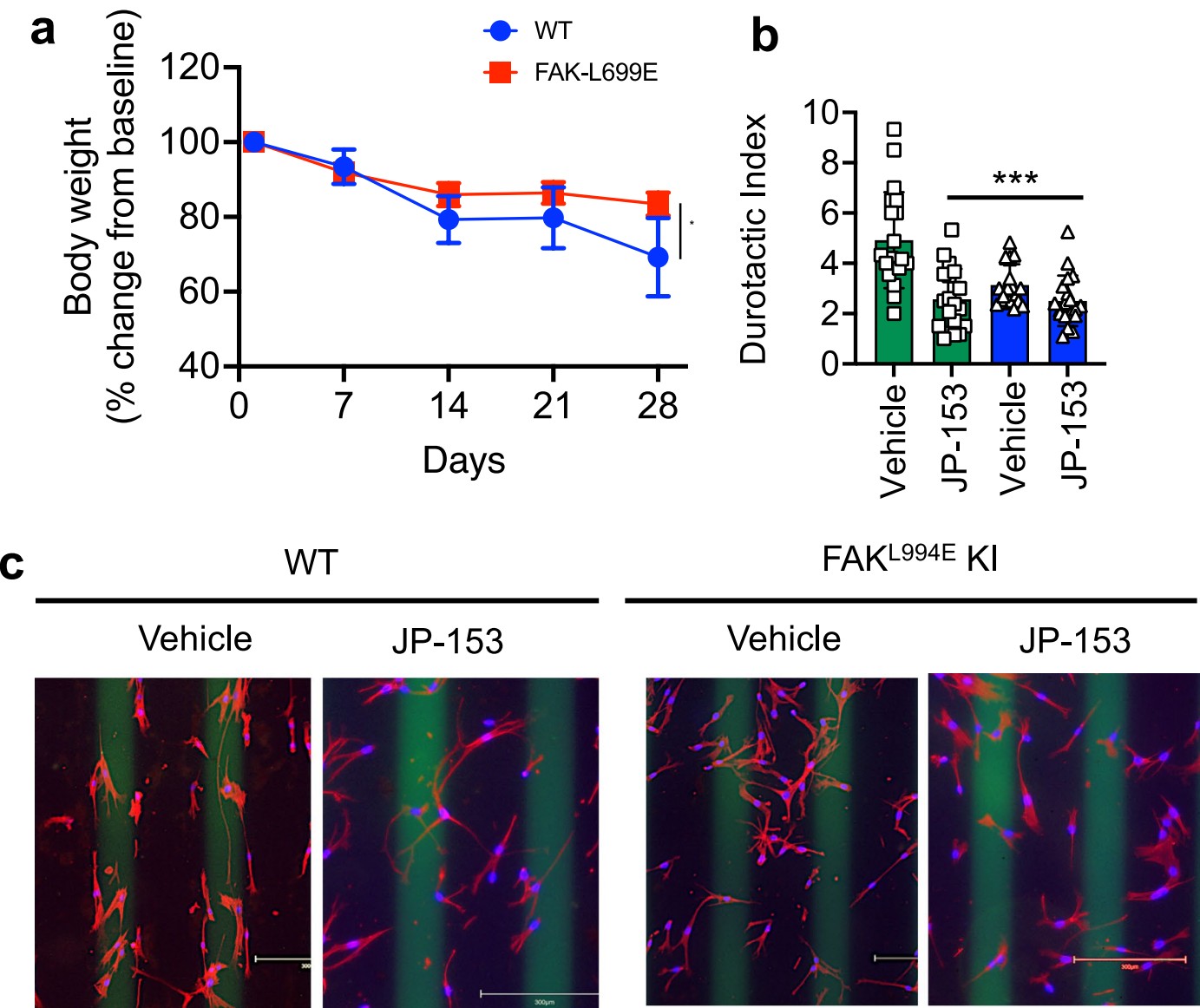

**Extended Data Fig. 5 | Genetic and pharmacological inhibition of the FAK-Paxillin pathway inhibits fibroblast durotaxis and fibrosis development *in vivo*.** (**a**) Body weight changes of bleomycin treated FAK[L994E] KI and WT mice as the disease progresses with n = 5 for each group. (**b**) Assessment of durotaxis in isolated primary fibroblasts from bleomycin treated FAK[L994E] KI and WT mice in the presence of absence of JP-153 with n = 3 for each group. (**c**) Representative images of isolated primary fibroblasts from bleomycin treated FAK[L994E] KI and WT mice in the presence of absence of JP-153 in durotaxis assay. Data are given as mean ± SD with n = 3 for each group. P value was determined by Student's t test.

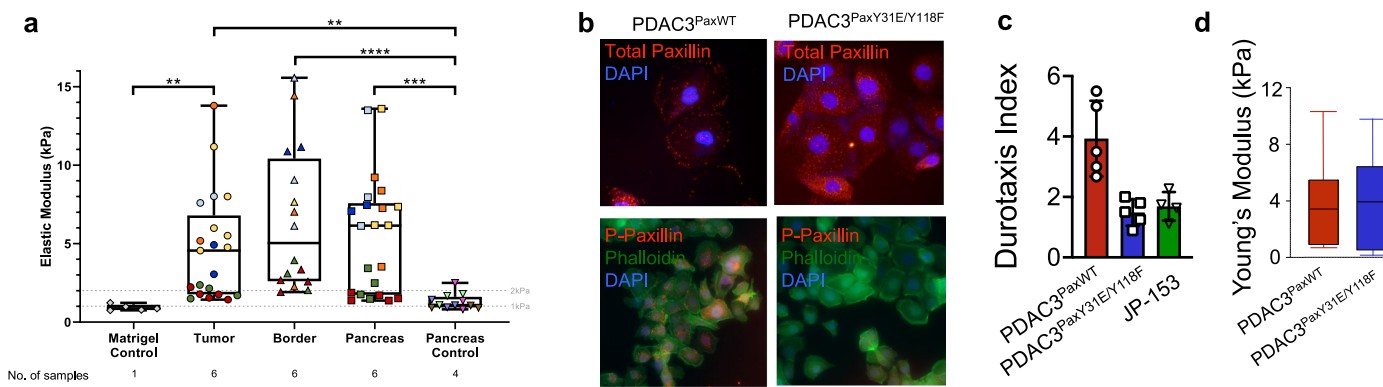

**Extended Data Fig. 6 | Genetic or pharmacological inhibition of the FAK-Paxillin pathway inhibits tumor cell durotaxis *in vitro* and *in vivo*.**
(**a**) Measurement of matrix stiffness in mouse pancreatic tissues by atomic force microscopy (AFM). AFM was applied to map local elastic properties of thin slices of fresh pancreatic samples prepared from the xenograft tumor model described in Fig. 5. Elastic modulus was measured in healthy pancreas, the primary pancreatic tumor, the tumor-tumor microenvironment border, and the tumor microenvironment (TME). Matrigel was used as control for mechanical measurements. Data are given as means ± SD with n = 6 for each group. P value was determined by Student's t test. (**b**) The wild type PDAC3 tumor cell line (herein PDAC3$^{PxnWT}$) was genetically engineered by transducing lentivirus encoding human Paxillin siRNA and mutated Y31E/Y118F chicken paxillin, effectively knocking down human paxillin while overexpressing mutated chicken

paxillin (herein PDAC3$^{PxnY31E/Y118F}$). Cells were stained with phalloidin (green) to visualize F-actin, total Paxillin and phospho-Paxillin Y31 (red) and 4′,6-diamidino-2-phenylindole (DAPI; blue) to visualize nuclei. Note the overexpression of total paxillin and lack of phospho-paxillin in the mutated PDAC3$^{PxnY31E/Y118F}$ line compared to PDAC3$^{PxnWT}$. Scale bar 20 µm. Representative images from 4 biological replicates. (**c**) Assessment of durotaxis in PDAC3$^{PxnY31E/Y118F}$, PDAC3$^{PxnWT}$, PDAC3$^{PxnWT}$ treated with JP-153 (300 nM) at 24 h. Data are given as mean ± SD with n = 5 for each group. P value was determined by Student's t test. (**d**) Measurement of matrix stiffness in mouse pancreatic tissues prepared from mice orthotopically injected with PDAC3$^{PxnWT}$ or PDAC3$^{PxnY31E/Y118F}$ tumor cell lines into the pancreas of SCID mice. Data are given as mean ± SD with n = 6 for all groups.

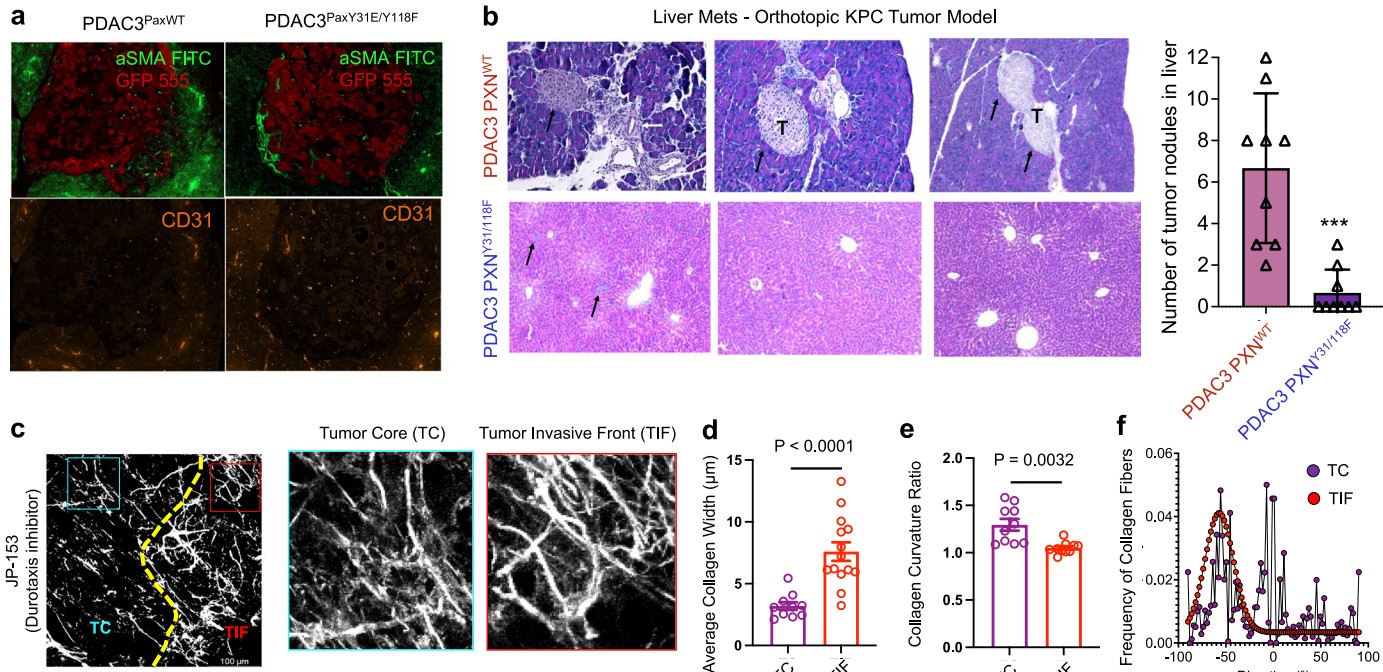

**Extended Data Fig. 7 | Effect of genetic inhibition of the FAK-Paxillin pathway on angiogenesis, liver metastases and collagen fiber structure *in vivo*.** (**a**) Immunofluorescence showing tumor cell growth (GFP+ cells, red), myofibroblast formation (α-SMA+ cells, green), and angiogenesis (CD31+ cells, orange) in mouse pancreatic tissues prepared from mice orthotopically injected with PDAC3^PxnWT or PDAC3^PxnY31E/Y118F tumor cell lines into the pancreas of SCID mice. Representative images with n = 6 for each group. Scale bar 100 μm. (**b**) Histological analysis showing the higher number of liver metastasis in mice

with PDAC3^PxnWT tumor compared to PDAC3^PxnY31E/Y118F tumor. Representative SHG images of tumor slices post-treatment with JP-153. (**a**) SHG images of the collagen network at the tumor core (TC) and tumor invasive front (TIF) of tumor slices. (**b**) Collagen fiber width distribution, (**c**) collagen fiber orientation distribution, and (**d**) collagen fiber curvature, defined by the curvature ratio, measured from SHG images at the TC and TIF areas. Data are given as mean ± SD with n = 6 for all groups.

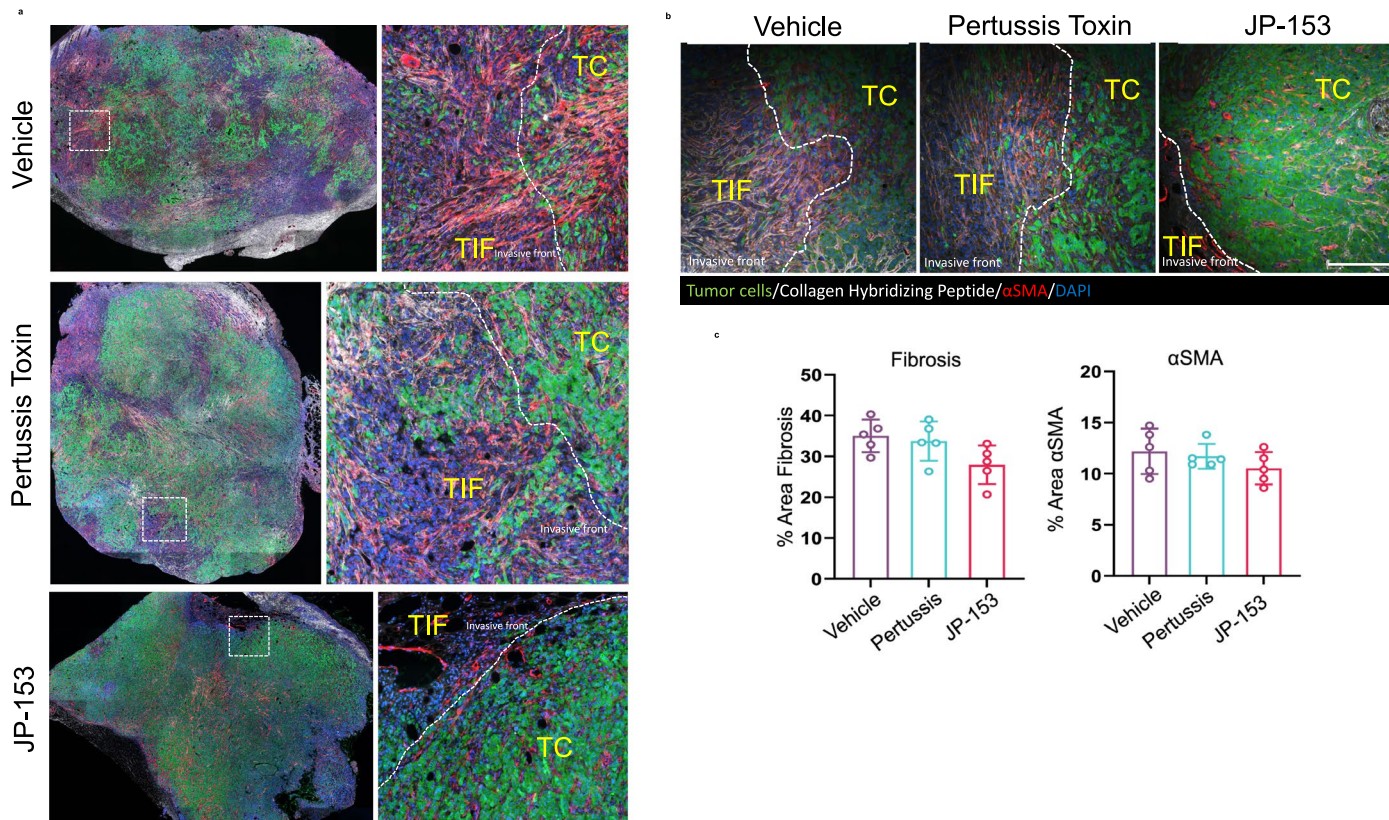

**Extended Data Fig. 8 | Effect of JP-153 on tumor cell invasion and fibrosis**
***in vivo.*** (**a**) Representative multiplex immunofluorescence images of tumor
slices post-treatment with vehicle, pertussis toxin, or JP-153. Staining for tumor
cells (green), α-SMA (red), collagen (white), and dapi (blue). (**b**) Staining for
tumor cells (green), α-SMA (red), collagen hybridizing peptide (white), and
dapi (blue). (**c**) Quantification of tumor fibrosis and α-SMA, shown as % Area.

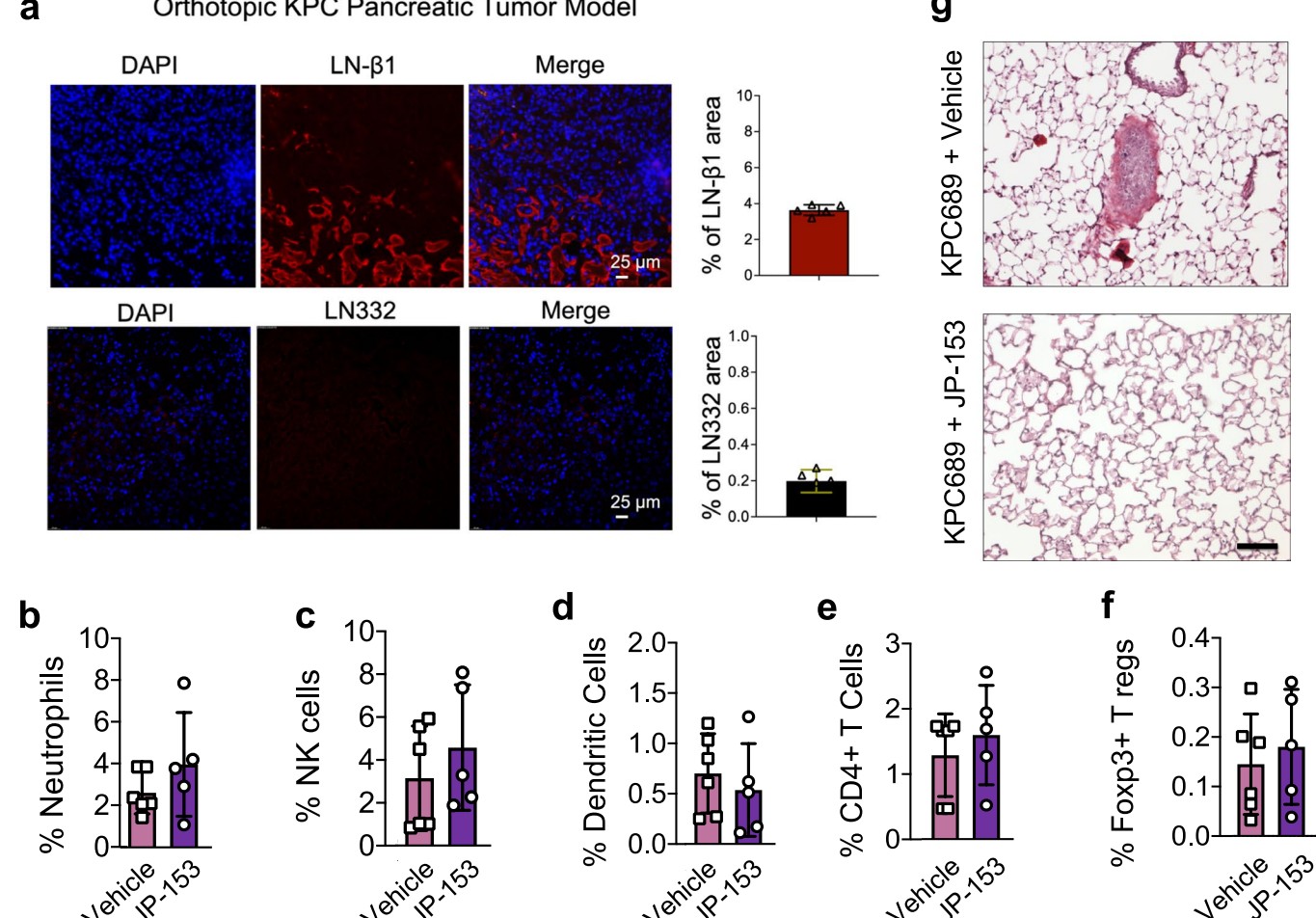

**Extended Data Fig. 9 | JP-153 has no effect on immune cells infiltration and changes in the basement membrane level.** (**a**) Immunofluorescence of laminin LN-β1, a component of nascent BMs, and LN-332, a component of mature BMs, early basement membrane marker in PDAC tumors. Data are given as mean ± SD with n = 6 for all groups. (**b-f**) Effect of JP-153 on the immune response in the primary tumor, assessed by flow cytometry (n = 6 for each group). Data are given as mean ± SD with n = 6 for all groups. (**g**) Lung metastases assessed by picrosirius red staining. Representative experiment with n = 6 for each group. Scale bar 100 µm.

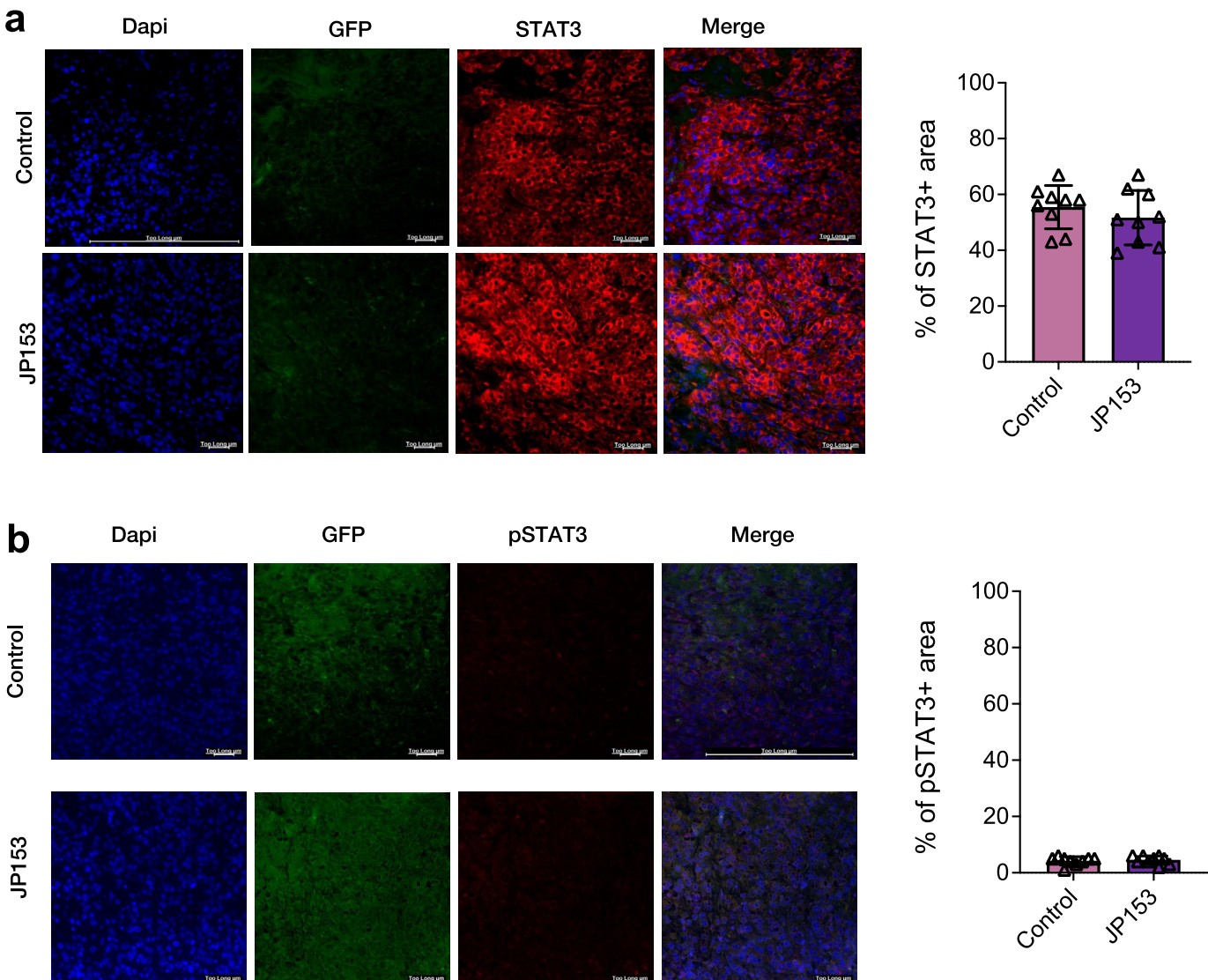

**Extended Data Fig. 10 | Effect of blocking FAK-Paxillin by JP-153 on tumor intrinsic STAT3 pathway.** Representative immunofluorescence staining for STAT3 (**a**) and pSTAT3 (**b**) in pancreatic tumor tissue after vehicle and JP-153 treatment. Please note that JP-153 did not affect the tumor intrinsic STAT3 signaling. Scale bar, 100 μm. Data are given as mean ± SD with n = 6 for all groups.

# Reporting Summary

## Statistics

For all statistical analyses, confirm that the following items are present in the figure legend, table legend, main text, or Methods section.

| n/a | Confirmed | |
|---|---|---|
| ☐ | ☒ | The exact sample size (*n*) for each experimental group/condition, given as a discrete number and unit of measurement |
| ☐ | ☒ | A statement on whether measurements were taken from distinct samples or whether the same sample was measured repeatedly |
| ☐ | ☒ | The statistical test(s) used AND whether they are one- or two-sided *Only common tests should be described solely by name; describe more complex techniques in the Methods section.* |
| ☐ | ☒ | A description of all covariates tested |
| ☐ | ☒ | A description of any assumptions or corrections, such as tests of normality and adjustment for multiple comparisons |
| ☐ | ☒ | A full description of the statistical parameters including central tendency (e.g. means) or other basic estimates (e.g. regression coefficient) AND variation (e.g. standard deviation) or associated estimates of uncertainty (e.g. confidence intervals) |
| ☐ | ☒ | For null hypothesis testing, the test statistic (e.g. *F*, *t*, *r*) with confidence intervals, effect sizes, degrees of freedom and *P* value noted *Give P values as exact values whenever suitable.* |
| ☒ | ☐ | For Bayesian analysis, information on the choice of priors and Markov chain Monte Carlo settings |
| ☒ | ☐ | For hierarchical and complex designs, identification of the appropriate level for tests and full reporting of outcomes |
| ☒ | ☐ | Estimates of effect sizes (e.g. Cohen's *d*, Pearson's *r*), indicating how they were calculated |

*Our web collection on statistics for biologists contains articles on many of the points above.*

## Software and code

Policy information about availability of computer code

| Data collection | Only commercial software, included with the instruments described in the methods, was used in data collection: AFM data was acquired using Bio-Catalyst AFM and MFP-3D AFM. Imaging of the fresh mouse tumor slices was conducted using a Leica SP8-MP upright multiphoton microscope with a Coherent Chameleon Vision II MP laser, equipped. Images were acquired with a Zeiss LSM780 confocal microscope (Zeiss). Flow cytometry was performed using a BDLSR Fortessa X-20 cell analyzer. IVIS imaging was use to visualize tumors in the pancreas, skin, livers and lungs. |
|---|---|
| Data analysis | Data was analyzed and visualized using Microsoft Excel and GraphPad Prism 5.0. DNA and protein sequences were analyzed using Geneious. Flow cytometry data was analyzed using FloJo. Images and time lapse videos were made and/or analyzed using FIJI, NIH ImageJ, MetaMorph 6.1, Manual Tracking Plugin and Correct 3D Drift Plugin. Two Photon Imaging Videos were generated by Imaris and LAS X software. Collagen measurement was conducted using CT-FIRE software. AFM data was analyzed using Bio-Catalyst AFM and MFP-3D AFM integrated softwares, resulting young modulus data was plotted in 3D stiffness maps using MATLAB. |

For manuscripts utilizing custom algorithms or software that are central to the research but not yet described in published literature, software must be made available to editors and reviewers. We strongly encourage code deposition in a community repository (e.g. GitHub). See the Nature Portfolio guidelines for submitting code & software for further information.

## Data

Policy information about availability of data

All manuscripts must include a data availability statement. This statement should provide the following information, where applicable:
  - Accession codes, unique identifiers, or web links for publicly available datasets
  - A description of any restrictions on data availability
  - For clinical datasets or third party data, please ensure that the statement adheres to our policy

The authors declare that all data generated or analyzed during this study are available upon request. Requests for raw or additional data should be emailed to the corresponding author and should include a brief description of the proposed analysis. Requests for data access will be reviewed individually, and a decision will be communicated within 4 weeks of receipt. Patient-derived data containing confidential or identifiable patient information are subject to patient privacy and cannot be shared. All data is available upon request.

## Research involving human participants, their data, or biological material

Policy information about studies with human participants or human data. See also policy information about sex, gender (identity/presentation), and sexual orientation and race, ethnicity and racism.

| | |
|---|---|
| Reporting on sex and gender | N/A |
| Reporting on race, ethnicity, or other socially relevant groupings | N/A |
| Population characteristics | N/A |
| Recruitment | N/A |
| Ethics oversight | All human experiments were performed under protocols approved by the Institutional Ethics Committee approved by the Massachusetts General Hospital. Patients with IPF were identified from those receiving care at the Massachusetts General Hospital. For study inclusion, patients with IPF had to satisfy IPF diagnostic criteria based on the 2011 joint consensus statement of the American Thoracic Society (ATS), European Respiratory Society (ERS), Japanese Respiratory Society, and Latin American Thoracic Association as determined by 2 investigators. |

Note that full information on the approval of the study protocol must also be provided in the manuscript.

# Field-specific reporting

Please select the one below that is the best fit for your research. If you are not sure, read the appropriate sections before making your selection.

☒ Life sciences    ☐ Behavioural & social sciences    ☐ Ecological, evolutionary & environmental sciences

For a reference copy of the document with all sections, see nature.com/documents/nr-reporting-summary-flat.pdf

# Life sciences study design

All studies must disclose on these points even when the disclosure is negative.

| | |
|---|---|
| Sample size | In each in vivo animal experiment evaluating the effects of genetic inhibition of the FAK-Paxillin pathway on the extent of dermal and lung fibrosis produced in mice, we use > or = 8 mice per group to achieve statistical significance and account for the inherent variability in the fibrotic response of mice. Assuming a 50% reduction in the amount of fibrosis present in FAK-Paxillin Knock-In mice compared with wild-type control mice, then at least 8 mice per group were needed to achieve a power of 80%, accepting a Type I error rate of 0.05. Histological analyses were done using n= 5 mice per group. Collagen determinations by hydroxyproline levels were performed using n=5 mice per group.

In each fibrosis experiment evaluating the effects of JP-153 inhibitor on the extent of lung fibrosis produced in mice, we use > or = 10 mice per group to achieve statistical significance and account for the inherent variability in the fibrotic response of mice. Assuming a 50% reduction in the amount of fibrosis present in mice treated with active drug compared with vehicle control, then 10 mice per group were needed to achieve a power of 80%, accepting a Type I error rate of 0.05. Collagen determinations by hydroxyproline levels were performed using n=6 mice per group, flow cytometry studies were performed using n= 4-6, histological analysis of lung tissues were done using n=6 mice per group. Western blot analyses from our in vivo mouse model of lung fibrosis includes 6 samples per condition. Representative blot shows at least n=3 samples/group.

In each tumor experiment evaluating the effects of genetic inhibition of FAK-Paxillin or pharmacological inhibition of the pathway with JP-153 inhibitor on tumor growth, tumor fibrosis and metastasis. Power analysis was used to determine the sample size of at least eight mice per group (80% power for an effect size of at least 1.5, assuming 5% significance level and a two-sided test). |
| Data exclusions | No data was excluded. |

| | | | |
|---|---|---|---|
| Replication | Number of replicates is described in the figure legends, where applicable. All attempts of replication were successful. |
| Randomization | For genetic or pharmacological experiments with JP-153 inhibitor, all mice used were wild type (C57Bl/6N) animals, and were purchased commercially and randomized to experimental groups by the cages our animal facility put them in upon their arrival from the vendor. |
| Blinding | The investigators were blinded to group allocation during data collection and analyses. |

# Reporting for specific materials, systems and methods

We require information from authors about some types of materials, experimental systems and methods used in many studies. Here, indicate whether each material, system or method listed is relevant to your study. If you are not sure if a list item applies to your research, read the appropriate section before selecting a response.

## Materials & experimental systems

| n/a | Involved in the study |
|---|---|
| ☐ | ☒ Antibodies |
| ☐ | ☒ Eukaryotic cell lines |
| ☒ | ☐ Palaeontology and archaeology |
| ☐ | ☒ Animals and other organisms |
| ☒ | ☐ Clinical data |
| ☒ | ☐ Dual use research of concern |
| ☒ | ☐ Plants |

## Methods

| n/a | Involved in the study |
|---|---|
| ☒ | ☐ ChIP-seq |
| ☐ | ☒ Flow cytometry |
| ☒ | ☐ MRI-based neuroimaging |

## Antibodies

| | |
|---|---|
| Antibodies used | Antibodies used were as follows: α-SMA (1A4, Sigma-Aldrich); Phospho-FAK (Tyr397) (#3283, Cell Signaling), FAK (#3285, Cell Signaling), Phospho-Paxillin (Tyr118) (MAB61641, R&D), Phospho-Paxillin (Tyr31) (#2541, Cell Signaling), Paxillin (MA124952, Invitrogen), Chicken Paxillin (Clone: PXC-10, Invitrogen), GFP (#2555, Cell Signaling), YAP (H-9, sc-271134, Santa Cruz Biotechnologies), CD31 (PECAM-1) (#77699, Cell Signaling), STAT3 (12640, Cell Signaling), Phospho-Stat3 (Tyr705) (D3A7) (9145, Cell Signaling) laminin-β1 (LT3, sc-33709, Santa Cruz), laminin-332 (711306, Invitrogen), β-actin (#4970, Cell Signaling), GAPDH (glyceraldehyde-3-phosphate dehydrogenase) (Cell Signaling). Secondary antibodies were obtained from Invitrogen [Alexa Fluor 488 goat anti-mouse immunoglobulin G2a (IgG2a) and Alexa Fluor 555 goat anti-rabbit IgG1]. F-actin and nuclei were stained with Alexa Fluor 546–phalloidin and 4′,6-diamidino-2-phenylindole (DAPI) (Invitrogen), respectively. Antibody dilutions were prepared according to the manufacturer's guidelines |
| Validation | Antibodies were used within the uses validated by the manufacturers. |

## Eukaryotic cell lines

Policy information about cell lines and Sex and Gender in Research

| | |
|---|---|
| Cell line source(s) | Mesenchymal stem cells (MSCs, Lonza), human lung fibroblasts (IMR-90, ATCC, CCL-186), human foreskin fibroblasts (ATCC, SCRC-1041), primary normal dermal fibroblasts (ATCC, PCS-201-010), human primary kidney fibroblasts (Cell Biologics, H-6016), human primary liver fibroblasts (ATCC, FL 62891), primary umbilical vein endothelial cells (HUVEC, ATCC, PCS-100-013), human embryonic kidney cells (293T, ATCC, CRL-1573) were purchased from commercial vendors. Primary lung mouse fibroblasts, mouse leukocytes, and mouse lung endothelial cells (MLEC) were isolated from C57Bl6 mice by tissue-digestion process. Healthy lung fibroblasts were isolated from lung sections from patients without IPF that underwent lung transplant. PDAC 2, PDAC 3, PDAC 9, PDAC 5, PDAC6 and PDAC8 tumor cell lines and primary cancer associated fibroblast (CAF) line were kindly provided by Drs. David Ting and Matteo Ligorio (MGH Cancer Center and Harvard Medical School, Boston, USA). These patient-derived PDAC cell lines were derived from metastatic ascites from patients under a discarded tissue protocol in accordance with the Massachusetts General Hospital (MGH) IRB protocol 2011P001236. Tumor cell lines were immortalized, constitutively expressing a GFP-Luciferase construct (74. CAFs were similarly immortalized for continual culturing by infecting with hTERT (pBABE-hygro-hTERT). The KPC689 cancer cell line was kindly provided by Dr. Raghu Kalluri (MD Anderson Cancer Center, Houston, Texas, USA) and established from the pancreatic tumors of Pdx1cre/+;LSL-KRasG12D/+;LSL-Trp53R172H/+ (KPC) mice. |
| Authentication | Commercial lines were validated by the vendors. No authentication was performed on primary cell cultures beyond positive/negative selection methods. |
| Mycoplasma contamination | All cells used were tested negative for mycoplasma. Mycoplasma testing is performed quarterly. |
| Commonly misidentified lines (See ICLAC register) | No commonly misidentified lines were used. |

# Animals and other research organisms

Policy information about studies involving animals; ARRIVE guidelines recommended for reporting animal research, and Sex and Gender in Research

| | |
|---|---|
| Laboratory animals | Pathogen-free male C57BL/6N (6- to 8-week-old) mice purchased from the National Cancer Institute (NCI) Frederick Mouse Repository were used for mouse models of skin, lung and kidney fibrosis as well syngeneic tumor models. Immunocompromised NOD/SCID/gamma-c (NSG; NOD.Cg-Prkdcscid Il2rgtm1Wjl/Sz, 6- to 8-week-old) obtained from Jackson Laboratories were housed used for the orthotopic xenograft model. |
| Wild animals | This study did not involve wild animals. |
| Reporting on sex | We used mainly male mice for fibrosis animal experiments because males have higher susceptibility to bleomycin injury and develop more severe fibrosis in both skin and lungs, compared with female mice. For tumor xenograft experiments, all mice were female and 4-6 weeks old. |
| Field-collected samples | *For laboratory work with field-collected samples, describe all relevant parameters such as housing, maintenance, temperature, photoperiod and end-of-experiment protocol OR state that the study did not involve samples collected from the field.* |
| Ethics oversight | All experiments were performed in accordance with National Institute of Health guidelines and protocols were approved by the Massachusetts General Hospital Subcommittee on Research Animal Care.All mice were maintained in a specific pathogen–free (SPF) environment certified by the American Association for Accreditation of Laboratory Animal Care (AAALAC). |

Note that full information on the approval of the study protocol must also be provided in the manuscript.

# Plants

| | |
|---|---|
| Seed stocks | *Report on the source of all seed stocks or other plant material used. If applicable, state the seed stock centre and catalogue number. If plant specimens were collected from the field, describe the collection location, date and sampling procedures.* |
| Novel plant genotypes | *Describe the methods by which all novel plant genotypes were produced. This includes those generated by transgenic approaches, gene editing, chemical/radiation-based mutagenesis and hybridization. For transgenic lines, describe the transformation method, the number of independent lines analyzed and the generation upon which experiments were performed. For gene-edited lines, describe the editor used, the endogenous sequence targeted for editing, the targeting guide RNA sequence (if applicable) and how the editor was applied.* |
| Authentication | *Describe any authentication procedures for each seed stock used or novel genotype generated. Describe any experiments used to assess the effect of a mutation and, where applicable, how potential secondary effects (e.g. second site T-DNA insertions, mosiacism, off-target gene editing) were examined.* |

# Flow Cytometry

## Plots

Confirm that:

☒ The axis labels state the marker and fluorochrome used (e.g. CD4-FITC).

☒ The axis scales are clearly visible. Include numbers along axes only for bottom left plot of group (a 'group' is an analysis of identical markers).

☒ All plots are contour plots with outliers or pseudocolor plots.

☒ A numerical value for number of cells or percentage (with statistics) is provided.

## Methodology

| | |
|---|---|
| Sample preparation | Single-cell suspensions were isolated from mouse lung tissues biopsies using Liberase Blendzyme (final concentration, 0.14U/ml; Roche) and deoxyribonuclease I (final concentration, 60 mg/ml; Sigma) for 45 min at 37°C.  Cells were incubated with FcRII and FcRIII blocking antibody (BioLegend, clone 93) for 10 min at 4°C followed by staining with the following fluorophore-conjugated antibody from Biolegend: Viability eF780 (1:1000), CD11b-BUV737 (1:100), Ly6G-FITC (1:200), Ly6C-PerCP-Cy5.5 (1:200), CCR2-PE (1:50), CD11c-BV605 (1:200), MHCII-Pe-Cy7 (1:1000), F4/80-PE (1:100), MerTK-APC (1:100), CD3-BUV395 (1:200), CD4-BV786 (1:200), CD8-FITC (1:200). |
| Instrument | Flow cytometry was performed using a BDLSR Fortessa X-20 cell analyzer |
| Software | FlowJo software V10 was used for analysis. |
| Cell population abundance | At least 10.000 cells were analyzed for each sample |

Gating strategy | Initial cell population gating (SSC vs FSC) was adopted to exclude the debris. A figure exemplifying the gating strategy is provide in the supplementary information.

☒ Tick this box to confirm that a figure exemplifying the gating strategy is provided in the Supplementary Information.

