## [Peer Review File · Nature Cell Biology]

Durotaxis is a driver and potential therapeutic target in lung fibrosis and metastatic pancreatic cancer

Corresponding Author: Professor David Lagares

This manuscript has been previously reviewed at another journal. This document only contains information relating to versions considered at Nature Cell Biology.

Version 0:

Decision Letter:

Our ref: NCB-A56458-T

5th March 2025

Dear Dr. Lagares,

Thank you for submitting your revised manuscript "Durotaxis is a driver and therapeutic target in organ fibrosis and metastatic cancer" (NCB-A56458-T). It has now been seen by the original referees and their comments are below. The reviewers find that the paper has improved in revision, and therefore we'll be happy in principle to publish it in Nature Cell Biology, pending minor revisions to satisfy the referees' final requests and to comply with our editorial and formatting guidelines.

Thank you again for your interest in Nature Cell Biology Please do not hesitate to contact me if you have any questions.

Sincerely,

Zhe Wang, PhD
Senior Editor
Nature Cell Biology

Tel: +44 (0) 207 843 4924
email: zhe.wang@nature.com

Reviewer #4 (Remarks to the Author):

The authors have significantly improved the manuscript by thoroughly addressing all the concerns raised in the reviews. They have also strengthened the discussion by acknowledging the limitations of their results. The manuscript is well written and clearly presented. This work will be of significant interest to researchers working at the intersection of biophysics, cancer and biology.

Reviewer #5 (Remarks to the Author):

Thank you for your continued effort to improve the manuscript. I appreciate your response to the comments I had last time

and except for my one continued hesitation I am supportive of this manuscript. I think it's both exciting and novel.

Unfortunately, I am still hesitant with the lack of biochemical validation of the JP compound. I appreciate the controls included (PLA and western blotting) as well as the newly added data comparing the FAK FAT mutation with the JP compound. I also appreciate the need to get this highly interesting study published, so I would suggest simply toning down the language used to describe the inhibitor fx in the abstract it says: 'selective inhibition of with the small molecule JP-153'. Here you can remove 'selective' as it's not clear from the data provided that the compound is selective. Secondly, in the discussion you describe JP-153 as a 'first in class durtaxis targeting drug' which again, without additional biochemical validation I think is an overstatement. Otherwise I am very supportive.

Version 1:

Decision Letter:

Dear Dr Lagares,

I am pleased to inform you that your manuscript, "Durtaxis is a driver and potential therapeutic target in lung fibrosis and metastatic pancreatic cancer", has now been accepted for publication in Nature Cell Biology.

Please note that *Nature Cell Biology* is a Transformative Journal (TJ). Authors may publish their research with us through the traditional subscription access route or make their paper immediately open access through payment of an article-processing charge (APC). Authors will not be required to make a final decision about access to their article until it has been accepted. <https://www.springernature.com/gp/open-research/transformative-journals> Find out more about Transformative Journals

Authors may need to take specific actions to achieve [compliance](https://www.springernature.com/gp/open-research/funding/policy-compliance-faqs) with funder and institutional open access mandates. If your research is supported by a funder that requires immediate open access (e.g. according to [compliance](https://www.springernature.com/gp/open-research/funding/policy-compliance-faqs))

[Plan S principles](https://www.springernature.com/gp/open-research/plan-s-compliance)) then you should select the gold OA route, and we will direct you to the compliant route where possible. For authors selecting the subscription publication route, the journal's standard licensing terms will need to be accepted, including [self-archiving policies](https://www.springernature.com/gp/open-research/policies/journal-policies). Those licensing terms will supersede any other terms that the author or any third party may assert apply to any version of the manuscript.

If you have not already done so, we strongly recommend that you upload the step-by-step protocols used in this manuscript to protocols.io (<https://protocols.io>), an open online resource that allows researchers to share their detailed experimental know-how. All uploaded protocols are made freely available and are assigned DOIs for ease of citation. Protocols and Nature Portfolio journal papers in which they are used can be linked to one another, and this link is clearly and prominently visible in the online versions of both. Authors who performed the specific experiments can act as primary authors for the Protocol as they will be best placed to share the methodology details, but the Corresponding Author of the present research paper should be included as one of the authors. By uploading your Protocols onto protocols.io, you are enabling researchers to more readily reproduce or adapt the methodology you use, as well as increasing the visibility of your protocols and papers. You can also establish a dedicated workspace to collect your lab Protocols. Further information can be found at <https://www.protocols.io/help/publish-articles>.

Nature Cell Biology encourages authors presenting evidence for cell, biological, molecular, and genetic interactions to consider communicating these findings using Biofactoid (<https://biofactoid.org/>). This tool helps users share a searchable representation of interactions (e.g. binding, gene expression, post-translational modification) between genes, gene products, or chemicals. Information added to Biofactoid, with author attribution, is shared on social media and public databases, such as Pathway Commons, where it can be discovered and analyzed in the context of a large and growing corpus of knowledge.

With kind regards,

Zhe Wang, PhD
Senior Editor
Nature Cell Biology

Tel: +44 (0) 207 843 4924
email: zhe.wang@nature.com

** Visit the Springer Nature Editorial and Publishing website at http://editorial-jobs.springernature.com?utm_source=ejp_NCB_email&utm_medium=ejp_NCB_email&utm_campaign=ejp_NCB for more information about our career opportunities. If you have any questions please click [here](mailto:editorial.publishing.jobs@springernature.com).
